# TOWARD GUIDANCE-FREE AR VISUAL GENERATION VIA CONDITION CONTRASTIVE ALIGNMENT

**Huayu Chen[1], Hang Su[1], Peize Sun[2], Jun Zhu[1,3*]**

[1]Department of Computer Science & Technology, Institute for AI, BNRist Center,
Tsinghua-Bosch Joint ML Center, THBI Lab, Tsinghua University
[2]The University of Hong Kong
[3]Shengshu Technology, Beijing

## ABSTRACT

Classifier-Free Guidance (CFG) is a critical technique for enhancing the sample quality of visual generative models. However, in autoregressive (AR) multi-modal generation, CFG introduces design inconsistencies between language and visual content, contradicting the design philosophy of unifying different modalities for visual AR. Motivated by language model alignment methods, we propose *Condition Contrastive Alignment* (CCA) to facilitate guidance-free AR visual generation with high performance and analyzes its theoretical connection with guided sampling methods. Unlike guidance methods that alter the sampling process to achieve the ideal sampling distribution, CCA directly fine-tunes pretrained models to fit the *same* distribution target. Experimental results show that CCA can significantly enhance the guidance-free performance of all tested models with just one epoch of fine-tuning (∼1% of pretraining epochs) on the pretraining dataset, on par with guided sampling methods. This largely removes the need for guided sampling in AR visual generation and cuts the sampling cost by half. Moreover, by adjusting training parameters, CCA can achieve trade-offs between sample diversity and fidelity similar to CFG. This experimentally confirms the strong theoretical connection between language-targeted alignment and visual-targeted guidance methods, unifying two previously independent research fields. Code and models: `https://github.com/thu-ml/CCA`.

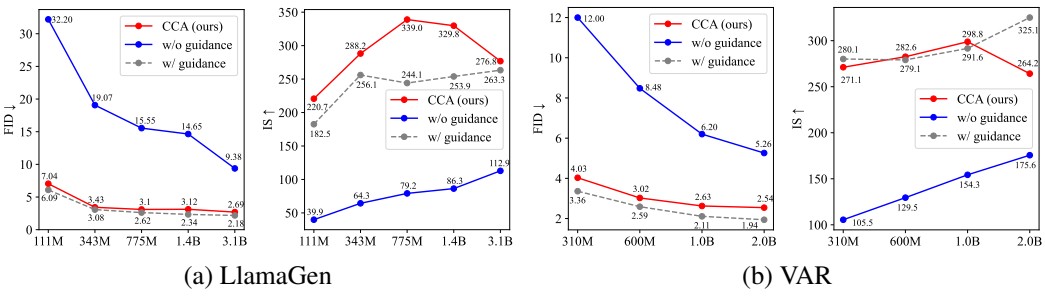

Figure 1: CCA significantly improves guidance-free sample quality for AR visual generative models with just one epoch of fine-tuning on the pretraining dataset.

## 1 INTRODUCTION

Witnessing the scalability and generalizability of autoregressive (AR) models in language domains, recent works have been striving to replicate similar success for visual generation (Esser et al., 2021; Lee et al., 2022). By quantizing images into discrete tokens, AR visual models can process images using the same *next-token prediction* approach as Large Language Models (LLMs). This approach is

---

*Corresponding author

attractive because it provides a potentially unified framework for vision and language, promoting consistency in reasoning and generation across modalities (Team, 2024; Xie et al., 2024).

Despite the design philosophy of maximally aligning visual modeling with language modeling methods, AR visual generation still differs from language generation in a notable aspect. AR visual generation relies heavily on Classifier-Free Guidance (CFG) (Ho & Salimans, 2022), a sampling technique unnecessary for language generation, which has caused design inconsistencies between the two types of content. During sampling, while CFG helps improve sample quality by contrasting conditional and unconditional models, it requires two model inferences per visual token, which doubles the sampling cost. During training, CFG requires randomly masking text conditions to learn the unconditional distribution, preventing the simultaneous training of text tokens (Team, 2024).

In contrast to visual generation, LLMs rarely rely on guided sampling. Instead, the surge of LLMs' instruction-following abilities has largely benefited from fine-tuning-based alignment methods (Schulman et al., 2022). Motivated by this observation, we seek to study: *"Can we avoid guided sampling in AR visual generation, but attain similar effects by directly fine-tuning pretrained models?"*

In this paper, we derive Condition Contrastive Alignment (CCA) for enhancing visual AR performance without guided sampling. Unlike CFG which necessitates altering the sampling process to achieve a more desirable sampling distribution, CCA directly fine-tunes pretrained AR models to fit the *same* distribution target, leaving the sampling scheme untouched. CCA is quite convenient to use since it does not rely on any additional datasets beyond the pretraining data. Our method functions by contrasting positive and negative conditions for a given image, which can be easily created from the existing pretraining dataset as matched or mismatched image-condition pairs. CCA is also highly efficient given its fine-tuning nature. We observe that our method achieves ideal performance within just one training epoch, indicating negligible computational overhead ($\sim$1% of pretraining).

In Sec. 4, we highlight a theoretical connection between CCA and guided sampling techniques (Dhariwal & Nichol, 2021; Ho & Salimans, 2022). Essentially these methods all target at the same sampling distribution. The distributional gap between this target distribution and pretrained models is related to a physical quantity termed conditional residual ($\log \frac{p(\boldsymbol{x}|\boldsymbol{c})}{p(\boldsymbol{x})}$). Guidance methods typically train an additional model (e.g., unconditional model or classifier) to estimate this quantity and enhance pretrained models by altering their sampling process. Contrastively, CCA follows LLM alignment techniques (Rafailov et al., 2023; Chen et al., 2024a) and parameterizes the conditional residual with the difference between our target model and the pretrained model, thereby directly training a sampling model. This analysis unifies language-targeted alignment and visual-targeted guidance methods, bridging the gap between the two previously independent research fields.

We apply CCA to two state-of-the-art autoregressive (AR) visual models, LLamaGen (Sun et al., 2024) and VAR (Tian et al., 2024), which feature distinctly different visual tokenization designs. Both quantitative and qualitative results show that CCA significantly and consistently enhances the guidance-free sampling quality across all tested models, achieving performance levels comparable to CFG (Figure 1). We further show that by varying training hyperparameters, CCA can realize a controllable trade-off between image diversity and fidelity similar to CFG. This further confirms their theoretical connections. We also compare our method with some existing LLM alignment methods (Welleck et al., 2019; Rafailov et al., 2023) to justify its algorithm design. Finally, we demonstrate that CCA can be combined with CFG to further improve performance.

Our contributions: 1. We take a big step toward guidance-free visual generation by significantly improving the visual quality of AR models. 2. We reveal a theoretical connection between alignment and guidance methods. This shows that language-targeted alignment can be similarly applied to visual generation and effectively replace guided sampling, closing the gap between these two fields.

## 2 BACKGROUND

### 2.1 AUTOREGRESSIVE (AR) VISUAL MODELS

**Autoregressive models.** Consider data $\boldsymbol{x}$ represented by a sequence of discrete tokens $\boldsymbol{x}_{1:N} := \{x_1, x_2, ..., x_N\}$, where each token $\boldsymbol{x}_n$ is an integer. Data probability $p(\boldsymbol{x})$ can be decomposed as:

$$p(\boldsymbol{x}) = p(\boldsymbol{x}_1) \prod_{n=2}^{N} p(\boldsymbol{x}_n | \boldsymbol{x}_{<n}). \tag{1}$$

AR models thus aim to learn $p_\phi(\boldsymbol{x}_n | \boldsymbol{x}_{<n}) \approx p(\boldsymbol{x}_n | \boldsymbol{x}_{<n})$, where each token $\boldsymbol{x}_n$ is conditioned only on its previous input $\boldsymbol{x}_{<n}$. This is known as *next-token prediction* (Radford et al., 2018).

**Visual tokenization.** Image pixels are continuous values, making it necessary to use vector-quantized tokenizers for applying discrete AR models to visual data (Van Den Oord et al., 2017; Esser et al., 2021). These tokenizers are trained to encode images $\boldsymbol{x}$ into discrete token sequences $\boldsymbol{x}_{1:N}$ and decode them back by minimizing reconstruction losses. In our work, we utilize pretrained and frozen visual tokenizers, allowing AR models to process images similarly to text.

### 2.2 GUIDED SAMPLING FOR VISUAL GENERATION

Despite the core motivation of developing a unified model for language and vision, the AR sampling strategies for visual and text contents differ in one key aspect: AR visual generation necessitates a sampling technique named Classifier-Free Guidance (CFG) (Ho & Salimans, 2022). During inference, CFG adjusts the sampling logits $\ell^{\text{sample}}$ for each token as:

$$\ell^{\text{sample}} = \ell^c + s(\ell^c - \ell^u), \tag{2}$$

where $\ell^c$ and $\ell^u$ are the conditional and unconditional logits provided by two separate AR models, $p_\phi(\boldsymbol{x}|\boldsymbol{c})$ and $p_\phi(\boldsymbol{x})$. The condition $\boldsymbol{c}$ can be class labels or text captions, formalized as prompt tokens. The scalar $s$ is termed guidance scale. Since token logits represent the (unnormalized) log-likelihood in AR models, Ho & Salimans (2022) prove that the sampling distribution satisfies:

$$p^{\text{sample}}(\boldsymbol{x}|\boldsymbol{c}) \propto p_\phi(\boldsymbol{x}|\boldsymbol{c}) \left[ \frac{p_\phi(\boldsymbol{x}|\boldsymbol{c})}{p_\phi(\boldsymbol{x})} \right]^s. \tag{3}$$

At $s = 0$, the sampling model becomes exactly the pretrained conditional model $p_\phi$. However, previous works (Ho & Salimans, 2022; Podell et al., 2023; Chang et al., 2023; Sun et al., 2024) have widely observed that an appropriate $s > 0$ is critical for an ideal trade-off between visual fidelity and diversity, making training another unconditional model $p_\phi$ necessary. In practice, the unconditional model usually shares parameters with the conditional one, and can be trained concurrently by randomly dropping condition prompts $\boldsymbol{c}$ during training.

Other guidance methods, such as Classifier Guidance (Ho & Salimans, 2022) and Energy Guidance (Lu et al., 2023) have similar effects of CFG. The target sampling distribution of these methods can all be unified under Eq. 3.

### 2.3 DIRECT PREFERENCE OPTIMIZATION FOR LANGUAGE MODEL ALIGNMENT

Reinforcement Learning from Human Feedback (RLHF) is crucial for enhancing the instruction-following ability of pretrained Language Models (LMs) (Schulman et al., 2022; OpenAI, 2023). Performing RL typically requires a reward model, which can be learned from human preference data. Formally, the Bradley-Terry preference model (Bradley & Terry, 1952) assumes.

$$p(\boldsymbol{x}_w \succ \boldsymbol{x}_l | \boldsymbol{c}) := \frac{e^{r(\boldsymbol{c}, \boldsymbol{x}_w)}}{e^{r(\boldsymbol{c}, \boldsymbol{x}_l)} + e^{r(\boldsymbol{c}, \boldsymbol{x}_w)}} = \sigma(r(\boldsymbol{c}, \boldsymbol{x}_w) - r(\boldsymbol{c}, \boldsymbol{x}_l)), \tag{4}$$

where $\boldsymbol{x}_w$ and $\boldsymbol{x}_l$ are respectively the winning and losing response for an instruction $\boldsymbol{c}$, evaluated by human. $r(\cdot)$ represents an implicit reward for each response. The target LM $\pi_\theta$ should satisfy $\pi_\theta(\boldsymbol{x}|\boldsymbol{c}) \propto \mu_\phi(\boldsymbol{x}|\boldsymbol{c}) e^{r(\boldsymbol{c}, \boldsymbol{x})/\beta}$ to attain higher implicit reward compared with the pretrained LM $\mu_\phi$.

Direct Preference Optimization (Rafailov et al., 2023) allows us to directly optimize pretrained LMs on preference data, by formalizing $r_\theta(\boldsymbol{c}, \boldsymbol{x}) := \beta \log \pi_\theta(\boldsymbol{x}|\boldsymbol{c}) - \beta \log \mu_\phi(\boldsymbol{x}|\boldsymbol{c})$:

$$\mathcal{L}_\theta^{\text{DPO}} = -\mathbb{E}_{\{\boldsymbol{c}, \boldsymbol{x}_w \succ \boldsymbol{x}_l\}} \log \sigma \left( \beta \log \frac{\pi_\theta(\boldsymbol{x}_w|\boldsymbol{c})}{\mu_\phi(\boldsymbol{x}_w|\boldsymbol{c})} - \beta \log \frac{\pi_\theta(\boldsymbol{x}_l|\boldsymbol{c})}{\mu_\phi(\boldsymbol{x}_l|\boldsymbol{c})} \right). \tag{5}$$

DPO is more streamlined and thus often more favorable compared with traditional two-stage RLHF pipelines: first training reward models, then aligning LMs with reward models using RL.

## 3 CONDITION CONTRASTIVE ALIGNMENT

Autoregressive visual models are essentially learning a parameterized model $p_\phi(\boldsymbol{x}|\boldsymbol{c})$ to approximate the standard conditional image distribution $p(\boldsymbol{x}|\boldsymbol{c})$. Guidance algorithms shift the sampling policy $p^{\text{sample}}(\boldsymbol{x}|\boldsymbol{c})$ away from $p(\boldsymbol{x}|\boldsymbol{c})$ according to Sec. 2.2:

$$p^{\text{sample}}(\boldsymbol{x}|\boldsymbol{c}) \propto p(\boldsymbol{x}|\boldsymbol{c}) \left[ \frac{p(\boldsymbol{x}|\boldsymbol{c})}{p(\boldsymbol{x})} \right]^s. \tag{6}$$

At guidance scale $s = 0$, sampling from $p^{\text{sample}}(\boldsymbol{x}|\boldsymbol{c}) = p(\boldsymbol{x}|\boldsymbol{c}) \approx p_\phi(\boldsymbol{x}|\boldsymbol{c})$ is most straightforward. However, it is widely observed that an appropriate $s > 0$ usually leads to significantly enhanced sample quality. The cost is that we rely on an extra unconditional model $p_\phi(\boldsymbol{x}) \approx p(\boldsymbol{x})$ for sampling. This doubles the sampling cost and causes an inconsistent training paradigm with language.

In this section, we derive a simple approach to directly model the same target distribution $p^{\text{sample}}$ using a **single** AR model $p_\theta^{\text{sample}}$. Specifically, our methods leverage a singular loss function for directly optimizing pretrained models $p_\phi(\boldsymbol{x}|\boldsymbol{c}) \approx p(\boldsymbol{x}|\boldsymbol{c})$ to become $p_\theta^{\text{sample}}(\boldsymbol{x}|\boldsymbol{c}) \approx p^{\text{sample}}(\boldsymbol{x}|\boldsymbol{c})$. Despite having similar effects as guided sampling, our approach does not require altering the sampling process. We theoretically derive our method in Sec. 3.1 and discuss its practical implementation in Sec. 3.2.

### 3.1 ALGORITHM DERIVATION

The core difficulty of directly learning $p_\theta^{\text{sample}}$ is that we cannot access datasets under the distribution of $p^{\text{sample}}$. However, we observe the distributional difference between $p^{\text{sample}}(\boldsymbol{x}|\boldsymbol{c})$ and $p(\boldsymbol{x}|\boldsymbol{c})$ is related to a simple quantity that can be potentially learned from existing datasets. Specifically, by taking the logarithm of both sides in Eq. 6 and applying some algebra, we have[1]:

$$\frac{1}{s} \log \frac{p^{\text{sample}}(\boldsymbol{x}|\boldsymbol{c})}{p(\boldsymbol{x}|\boldsymbol{c})} = \log \frac{p(\boldsymbol{x}|\boldsymbol{c})}{p(\boldsymbol{x})}, \tag{7}$$

of which the right-hand side (i.e., $\log \frac{p(\boldsymbol{x}|\boldsymbol{c})}{p(\boldsymbol{x})}$) corresponds to the log gap between the conditional probability and unconditional probability for an image $\boldsymbol{x}$, which we term as *conditional residual*.

Our key insight here is that the conditional residual can be directly learned through contrastive learning approaches (Gutmann & Hyvärinen, 2012), as sated below:

**Theorem 3.1** (Noise Contrastive Estimation, proof in Appendix A). *Let $r_\theta$ be a parameterized model which takes in an image-condition pair $(\boldsymbol{x}, \boldsymbol{c})$ and outputs a scalar value $r_\theta(\boldsymbol{x}, \boldsymbol{c})$. Consider the loss function:*

$$\mathcal{L}_\theta^{NCE}(\boldsymbol{x}, \boldsymbol{c}) = -\mathbb{E}_{p(\boldsymbol{x}, \boldsymbol{c})} \log \sigma(r_\theta(\boldsymbol{x}, \boldsymbol{c})) - \mathbb{E}_{p(\boldsymbol{x})p(\boldsymbol{c})} \log \sigma(-r_\theta(\boldsymbol{x}, \boldsymbol{c})), \tag{8}$$

*where $\sigma(\cdot)$ is the standard logistic function: $\sigma(w) := 1/(1 + e^{-w})$.*

*Given unlimited model expressivity for $r_\theta$, the optimal solution for minimizing $\mathcal{L}_\theta^{NCE}$ satisfies*

$$r_\theta^*(\boldsymbol{x}, \boldsymbol{c}) = \log \frac{p(\boldsymbol{x}|\boldsymbol{c})}{p(\boldsymbol{x})}. \tag{9}$$

Now that we have a tractable way of learning $r_\theta(\boldsymbol{x}, \boldsymbol{c}) \approx \log \frac{p(\boldsymbol{x}|\boldsymbol{c})}{p(\boldsymbol{x})}$, the target distribution $p^{\text{sample}}$ can be jointly defined by $r_\theta(\boldsymbol{x}, \boldsymbol{c})$ and the pretrained model $p_\phi$. However, we would still lack an explicitly

---

[1]We ignore a normalizing constant in Eq. 7 for brevity. A more detailed discussion is in Appendix B.

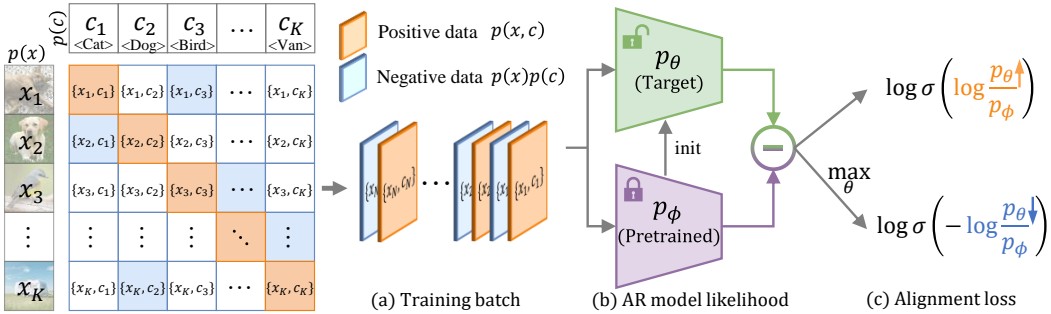

Figure 2: An overview of the CCA method. Given a training batch of $K$ <image, label> pairs, CCA treats these as positive samples, and generates $K$ negative samples by randomly assigning a negative label from $K-1$ remaining labels for each image. CCA then fine-tunes pretrained models by contrasting positive and negative data using an alignment loss. Pseudo code in Appendix D.

parameterized model $p_\theta^{\text{sample}}$ if $r_\theta(\boldsymbol{x}, \boldsymbol{c})$ is another independent network. To address this problem, we draw inspiration from the widely studied alignment techniques in language models (Rafailov et al., 2023) and parameterize $r_\theta(\boldsymbol{x}, \boldsymbol{c})$ with our target model $p_\theta^{\text{sample}}(\boldsymbol{x}|\boldsymbol{c})$ and $p_\phi(\boldsymbol{x}|\boldsymbol{c})$ according to Eq. 7:

$$r_\theta(\boldsymbol{x}, \boldsymbol{c}) := \frac{1}{s} \log \frac{p_\theta^{\text{sample}}(\boldsymbol{x}|\boldsymbol{c})}{p_\phi(\boldsymbol{x}|\boldsymbol{c})}. \tag{10}$$

Then, the loss function becomes

$$\mathcal{L}_\theta^{\text{CCA}} = -\mathbb{E}_{p(\boldsymbol{x}, \boldsymbol{c})} \log \sigma \left[ \frac{1}{s} \log \frac{p_\theta^{\text{sample}}(\boldsymbol{x}|\boldsymbol{c})}{p_\phi(\boldsymbol{x}|\boldsymbol{c})} \right] - \mathbb{E}_{p(\boldsymbol{x})p(\boldsymbol{c})} \log \sigma \left[ -\frac{1}{s} \log \frac{p_\theta^{\text{sample}}(\boldsymbol{x}|\boldsymbol{c})}{p_\phi(\boldsymbol{x}|\boldsymbol{c})} \right]. \tag{11}$$

During training, $p_\theta^{\text{sample}}$ is learnable while pretrained $p_\phi$ is frozen. $p_\theta^{\text{sample}}$ can be initialized from $p_\phi$.

This way we can fit $p^{\text{sample}}$ with a single AR model $p_\theta^{\text{sample}}$, eliminating the need for training a separate unconditional model for guided sampling. Sampling strategies for $p_\theta^{\text{sample}}$ are consistent with standard language model decoding methods, which unifies decoding systems for multi-modal generation.

## 3.2 PRACTICAL ALGORITHM

Figure 2 illustrates the CCA method. Specifically, implementing Eq. 11 requires approximating two expectations: one under the joint distribution $p(\boldsymbol{x}, \boldsymbol{c})$ and the other under the product of its two marginals $p(\boldsymbol{x})p(\boldsymbol{c})$. The key difference between these distributions is that in $p(\boldsymbol{x}, \boldsymbol{c})$, images $\boldsymbol{x}$ and conditions $\boldsymbol{c}$ are correctly paired. In contrast, $\boldsymbol{x}$ and $\boldsymbol{c}$ are sampled independently from $p(\boldsymbol{x})p(\boldsymbol{c})$, meaning they are most likely mismatched.

In practice, we rely solely on the pretraining dataset to estimate $\mathcal{L}_\theta^{\text{CCA}}$. Consider a batch of $K$ data pairs $\{\boldsymbol{x}, \boldsymbol{c}\}_{1:K}$. We randomly shuffle the condition batch $\boldsymbol{c}_{1:K}$ to become $\boldsymbol{c}_{1:K}^{\text{neg}}$, where each $\boldsymbol{c}_k^{\text{neg}}$ represents a negative condition of image $x_k$, while the original $\boldsymbol{c}_k$ is a positive one. This results in our training batch $\{\boldsymbol{x}, \boldsymbol{c}, \boldsymbol{c}^{\text{neg}}\}_{1:K}$. The loss function is

$$\mathcal{L}_\theta^{\text{CCA}}(\boldsymbol{x}_k, \boldsymbol{c}_k, \boldsymbol{c}_k^{\text{neg}}) = - \underbrace{\log \sigma \left[ \beta \log \frac{p_\theta^{\text{sample}}(\boldsymbol{x}_k|\boldsymbol{c}_k)}{p_\phi(\boldsymbol{x}_k|\boldsymbol{c}_k)} \right]}_{\text{relative likelihood for positive conditions} \uparrow} - \lambda \underbrace{\log \sigma \left[ -\beta \log \frac{p_\theta^{\text{sample}}(\boldsymbol{x}_k|\boldsymbol{c}_k^{\text{neg}})}{p_\phi(\boldsymbol{x}_k|\boldsymbol{c}_k^{\text{neg}})} \right]}_{\text{relative likelihood for negative conditions} \downarrow}, \tag{12}$$

where $\beta$ and $\lambda$ are two hyperparameters that can be adjusted. $\beta$ replaces the guidance scale parameter $s$, while $\lambda$ is for controlling the loss weight assigned to negative conditions. The learnable $p_\theta^{\text{sample}}$ is initialized from the pretrained conditional model $p_\phi$, making $\mathcal{L}_\theta^{\text{CCA}}$ a fine-tuning loss.

We give an intuitive understanding of Eq. 12. Note that $\log \sigma(\cdot)$ is monotonically increasing. The first term of Eq. 12 aims to increase the likelihood of an image $\boldsymbol{x}$ given a positive condition, with a

| Method | Classifier Guidance | Classifier-Free Guidance | Condition Contrastive Alignment |
|---|---|---|---|
| Modeling of $\log \frac{p(\boldsymbol{x}|\boldsymbol{c})}{p(\boldsymbol{x})}$ | $\log p_\theta(\boldsymbol{c}|\boldsymbol{x}) - \log p(\boldsymbol{c})$ | $\log p_\phi(\boldsymbol{x}|\boldsymbol{c}) - \log p_\theta(\boldsymbol{x})$ | $\beta[\log p_\theta^{\text{sample}}(\boldsymbol{x}|\boldsymbol{c}) - \log p_\phi(\boldsymbol{x}|c)]$ |
| Training loss | $\max_\theta \mathbb{E}_{p(\boldsymbol{x},\boldsymbol{c})} \log p_\theta(\boldsymbol{c}|\boldsymbol{x})$ | $\max_\theta \mathbb{E}_{p(\boldsymbol{x})} \log p_\theta(\boldsymbol{x})$ | $\min_\theta \mathcal{L}_\theta^{\text{CCA}}$ in Eq. 11 |
| Sampling policy | $\log p_\phi(\boldsymbol{x}|\boldsymbol{c}) + s \log p_\theta(\boldsymbol{c}|\boldsymbol{x})$ | $(1+s) \log p_\phi(\boldsymbol{x}|\boldsymbol{c}) - s \log p_\theta(\boldsymbol{x})$ | $\log p_\theta^{\text{sample}}(\boldsymbol{x}|\boldsymbol{c})$ |
| Extra training cost | $\sim$9% of learning $p_\phi$ | $\sim$10% of learning $p_\phi$ | $\sim$1% of pretraining $p_\phi$ |
| Sampling cost | $\times 1.3$ | $\times 2$ | $\times 1$ |
| Applicable area | Diffusion | Diffusion & Autoregressive | Autoregressive |

Table 1: Comparison of CCA (ours) and guidance methods in visual generative models.

similar effect to maximum likelihood training. For mismatched image-condition data, the second term explicitly minimizes its relative model likelihood compared with the pretrained $p_\phi$.

We name the above training technique Condition Contrastive Alignment (CCA) due to its contrastive nature in comparing positive and negative conditions with respect to a single image. This naming also reflects its theoretical connection with Noise Contrastive Estimation (Theorem 3.1).

## 4 CONNECTION BETWEEN CCA AND GUIDANCE METHODS

As summarized in Table 1, the key distinction between CCA and guidance methods is how to model $\log \frac{p(\boldsymbol{x}|\boldsymbol{c})}{p(\boldsymbol{x})}$, which defines the distributional gap between the target $p^{\text{sample}}(\boldsymbol{x}|\boldsymbol{c})$ and $p(\boldsymbol{x}|\boldsymbol{c})$ (Eq. 7).

In particular, Classifier Guidance (Dhariwal & Nichol, 2021) leverages Bayes' Rule and turn $\log \frac{p(\boldsymbol{x}|\boldsymbol{c})}{p(\boldsymbol{x})}$ into a conditional posterior:

$$\log \frac{p(\boldsymbol{x}|\boldsymbol{c})}{p(\boldsymbol{x})} = \log p(\boldsymbol{c}|\boldsymbol{x}) - \log p(\boldsymbol{c}) \approx \log p_\theta(\boldsymbol{c}|\boldsymbol{x}) - \log p(\boldsymbol{c}),$$

where $p(\boldsymbol{c}|\boldsymbol{x})$ is explicitly modeled by a classifier $p_\theta(\boldsymbol{c}|\boldsymbol{x})$, which is trained by a standard classification loss. $p(\boldsymbol{c})$ is regarded as a uniform distribution. CFG trains an extra unconditional model $p_\theta(\boldsymbol{x})$ to estimate the unknown part of $\log \frac{p(\boldsymbol{x}|\boldsymbol{c})}{p(\boldsymbol{x})}$:

$$\log \frac{p(\boldsymbol{x}|\boldsymbol{c})}{p(\boldsymbol{x})} \approx \log p_\phi(\boldsymbol{x}|\boldsymbol{c}) - \log p_\theta(\boldsymbol{x}).$$

Despite their effectiveness, guidance methods all require learning a separate model and a modified sampling process compared with standard autoregressive decoding. In comparison, CCA leverages Eq. 7 and models $\log \frac{p(\boldsymbol{x}|\boldsymbol{c})}{p(\boldsymbol{x})}$ as

$$\log \frac{p(\boldsymbol{x}|\boldsymbol{c})}{p(\boldsymbol{x})} \approx \beta[\log p_\theta^{\text{sample}}(\boldsymbol{x}|\boldsymbol{c}) - \log p_\phi(\boldsymbol{x}|c)],$$

which allows us to directly learn $p_\theta^{\text{sample}}$ instead of another guidance network.

Although CCA and conventional guidance techniques have distinct modeling methods, they all target at the same sampling distribution and thus have similar effects in visual generation. For instance, we show in Sec. 5.2 that CCA offers a similar trade-off between sample diversity and fidelity to CFG.

## 5 EXPERIMENTS

We seek to answer the following questions through our experiments:

1. How effective is CCA in enhancing the guidance-free generation quality of pretrained AR visual models, quantitatively and qualitatively? (Sec. 5.1)
2. Does CCA allow controllable trade-offs between sample diversity (FID) and fidelity (IS) similar to CFG? (Sec. 5.2)
3. How does CCA perform in comparison to alignment algorithms for LLMs? (Sec. 5.3)
4. Can CCA be combined with CFG to further improve performance? (Sec. 5.4)

| | Model | w/o Guidance | | | | w/ Guidance | |
|---|---|---|---|---|---|---|---|
| | | FID↓ | IS↑ | Precision↑ | Recall↑ | FID↓ | IS↑ |
| Diffusion | ADM  (Dhariwal & Nichol, 2021) | 7.49 | 127.5 | 0.72 | 0.63 | 3.94 | 215.8 |
| | LDM-4  (Rombach et al., 2022) | 10.56 | 103.5 | 0.71 | 0.62 | 3.60 | 247.7 |
| | U-ViT-H/2  (Bao et al., 2023) | – | – | – | – | 2.29 | 263.9 |
| | DiT-XL/2  (Peebles & Xie, 2023) | 9.62 | 121.5 | 0.67 | **0.67** | 2.27 | 278.2 |
| | MDTv2-XL/2  (Gao et al., 2023) | 5.06 | 155.6 | 0.72 | 0.66 | 1.58 | 314.7 |
| Mask | MaskGIT  (Chang et al., 2022) | 6.18 | 182.1 | 0.80 | 0.51 | – | – |
| | MAGVIT-v2  (Yu et al., 2023) | 3.65 | 200.5 | – | – | 1.78 | 319.4 |
| | MAGE  (Li et al., 2023) | 6.93 | 195.8 | – | – | – | – |
| Autoregressive | VQGAN  (Esser et al., 2021) | 15.78 | 74.3 | – | – | 5.20 | 280.3 |
| | ViT-VQGAN  (Yu et al., 2021) | 4.17 | 175.1 | – | – | 3.04 | 227.4 |
| | RQ-Transformer  (Lee et al., 2022) | 7.55 | 134.0 | – | – | 3.80 | 323.7 |
| | LlamaGen-3B  (Sun et al., 2024) | 9.38 | 112.9 | 0.69 | **0.67** | 2.18 | 263.3 |
| |   +CCA (Ours) | 2.69 | **276.8** | 0.80 | 0.59 | – | – |
| | VAR-d30  (Tian et al., 2024) | 5.25 | 175.6 | 0.75 | 0.62 | 1.92 | 323.1 |
| |   +CCA (Ours) | **2.54** | 264.2 | **0.83** | 0.56 | – | – |

Table 2: Model comparisons on class-conditional ImageNet $256 \times 256$ benchmark.

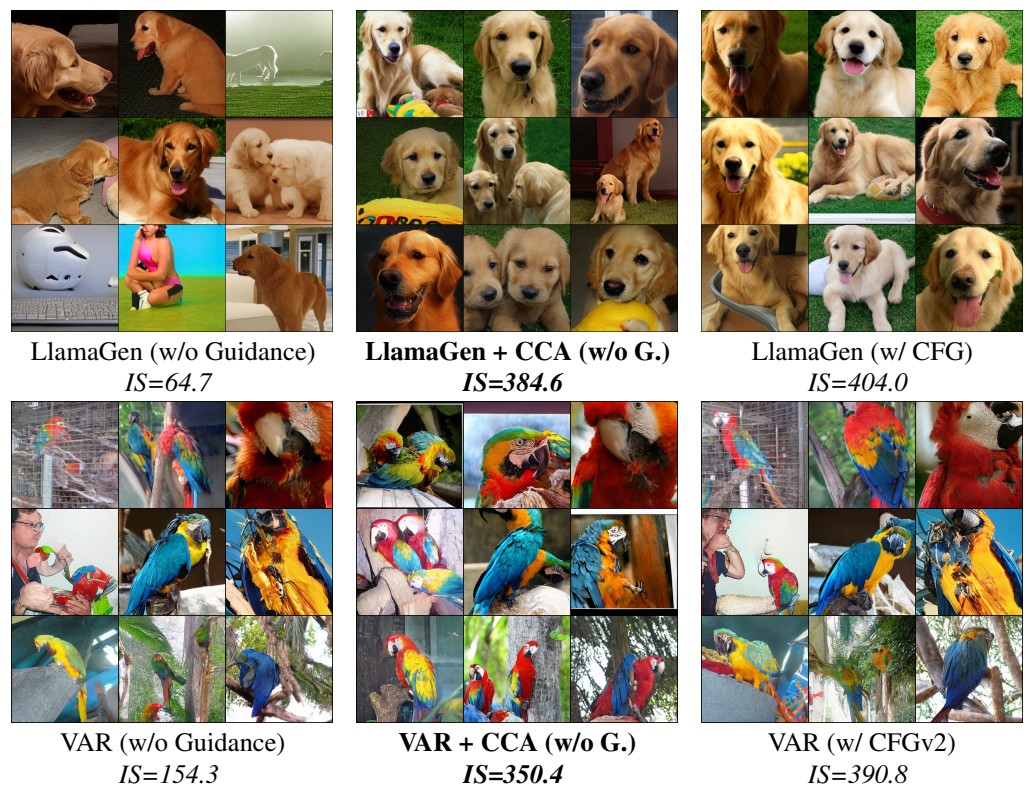

LlamaGen (w/o Guidance)
*IS=64.7*

**LlamaGen + CCA (w/o G.)**
*IS=384.6*

LlamaGen (w/ CFG)
*IS=404.0*

VAR (w/o Guidance)
*IS=154.3*

**VAR + CCA (w/o G.)**
*IS=350.4*

VAR (w/ CFGv2)
*IS=390.8*

Figure 3:  CCA and CFG can similarly enhance the sample fidelity of AR visual models. The base models are LlamaGen-L (343M) and VAR-d24 (1.0B). We use $s = 3.0$ for CFG, and $\beta = 0.02, \lambda = 10^4$ for CCA. Figure 7 and Figure 8 contain more examples.

## 5.1 TOWARD GUIDANCE-FREE AR VISUAL GENERATION

**Base model.**    We experiment on two families of publicly accessible AR visual models, LlamaGen (Sun et al., 2024) and VAR (Tian et al., 2024). Though both are class-conditioned models pretrained on ImageNet, LlamaGen and VAR feature distinctively different tokenizer and architecture designs.

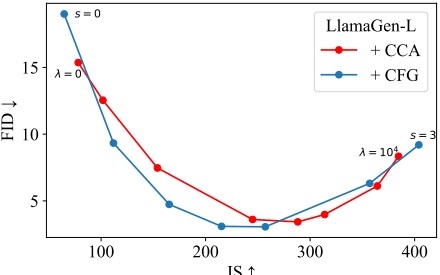 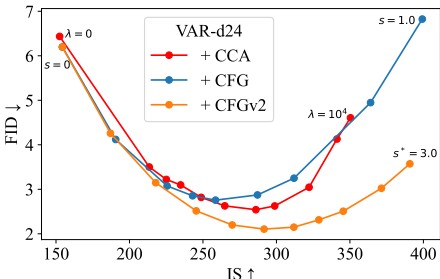

Figure 4: CCA can achieve similar FID-IS trade-offs to CFG by adjusting training parameter $\lambda$.

LlamaGen focuses on reducing the inductive biases on visual signals. It tokenizes images in the classic raster order and adopts almost the same LLM architecture as Llama (Touvron et al., 2023a). VAR, on the other hand, leverages the hierarchical structure of images and tokenizes them in a multi-scale, coarse-to-fine manner. VAR adopts a GPT-2 architecture but tailors the attention mechanism specifically for visual content. For both works, CFG is a default and critical technique.

**Training setup.** We leverage CCA to finetune multiple LlamaGen and VAR models of various sizes on the standard ImageNet dataset. The training scheme and hyperparameters are mostly consistent with the pretraining phase. We report performance numbers after only one training epoch and find this to be sufficient for ideal performance. We fix $\beta = 0.02$ in Eq. 12 and select suitable $\lambda$ for each model. Image resolutions are $384 \times 384$ for LlamaGen and $256 \times 256$ for VAR. Following the original work, we resize LlamaGen samples to $256 \times 256$ whenever required for evaluation.

**Experimental results.** We find CCA significantly improves the guidance-free performance of all tested models (Figure 1), evaluated by metrics like FID (Heusel et al., 2017) and IS (Salimans et al., 2016). For instance, after one epoch of CCA fine-tuning, the FID score of LlamaGen-L (343M) improves from 19.07 to 3.41, and the IS score from 64.3 to 288.2, achieving performance levels comparable to CFG. This result is compelling, considering that CCA has negligible fine-tunning overhead compared with model pretraining and only half of sampling costs compared with CFG.

Figure 3 presents a qualitative comparison of image samples before and after CCA fine-tuning. The results clearly demonstrate that CCA can vastly improve image fidelity, as well as class-image alignment of guidance-free samples.

Table 2 compares our best-performing models with several state-of-the-art visual generative models. With the help of CCA, we achieve a record-breaking FID score of 2.54 and an IS score of 276.8 for guidance-free samples of AR visual models. Although these numbers still somewhat lag behind CFG-enhanced performance, they demonstrate the significant potential of alignment algorithms to enhance visual generation and indicate the future possibility of replacing guided sampling.

## 5.2 Controllable Trade-Offs between Diversity and Fidelity

A distinctive feature of CFG is its ability to balance diversity and fidelity by adjusting the guidance scale. It is reasonable to expect that CCA can achieve a similar trade-off since it essentially targets the same sampling distribution as CFG.

Figure 4 confirms this expectation: by adjusting the $\lambda$ parameter for fine-tuning, CCA can achieve similar FID-IS trade-offs to CFG. The key difference is that CCA enhances guidance-free models through training, while CFG mainly improves the sampling process.

It is worth noting that VAR employs a slightly different guidance technique from standard CFG, which we refer to as CFGv2. CFGv2 involves linearly increasing the guidance scale $s$ during sampling, which was first proposed by Chang et al. (2023) and found beneficial for certain models. The FID-IS curve of CCA more closely resembles that of standard CFG. Additionally, the hyperparameter $\beta$ also affects CCA performance. Although our algorithm derivation shows that $\beta$ is directly related to the CFG scale $s$, we empirically find adjusting $\beta$ is less effective and less predictable compared with adjusting $\lambda$. In practice, we typically fix $\beta$ and adjust $\lambda$. We ablate $\beta$ in Appendix C.

| Model | FID↓ | IS | sFID↓ | Precision | Recall | Model | FID↓ | IS | sFID↓ | Precision | Recall |
|---|---|---|---|---|---|---|---|---|---|---|---|
| LlamaGen-L | 19.00 | 64.7 | 8.78 | 0.61 | **0.67** | VAR-d24 | 6.20 | 154.3 | 8.50 | 0.74 | **0.62** |
| +DPO | 61.69 | 30.8 | 44.98 | 0.36 | 0.40 | +DPO | 7.53 | 232.6 | 19.10 | **0.85** | 0.34 |
| +Unlearn | 12.22 | 111.6 | 7.99 | 0.66 | 0.64 | +Unlearn | 5.55 | 165.9 | 8.41 | 0.75 | 0.61 |
| **+CCA** | **3.43** | **288.2** | **7.44** | **0.81** | 0.52 | **+CCA** | **2.63** | **298.8** | **7.63** | 0.84 | 0.55 |

Table 3: Comparision of CCA and LLM alignment algorithms in visual generation.

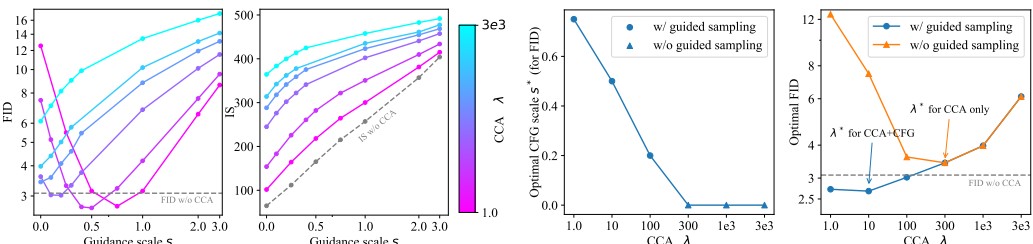

Figure 5: The impact of training parameter $\lambda$ on the performance of CCA+CFG.

## 5.3 Can LLM Alignment Algorithms also Enhance Visual AR?

Intuitively, existing preference-based LLM alignment algorithms such as DPO and Unlearning (Welleck et al., 2019) should also offer improvement for AR visual models given their similarity to CCA. However, Table 3 shows that naive applications of these methods fail significantly.

**DPO.** As is described in Eq. 5, one can treat negative image-condition pairs as dispreferred data and positive ones as preferred data to apply the DPO loss. We ablate $\beta_d \in \{0.01, 0.1, 1.0, 10.0\}$ and report the best performance in Table 3. Results indicate that DPO fails to enhance pretrained models, even causing performance collapse for LlamaGen-L. By inspecting training logs, we find that the likelihood of the positive data continuously decreases during fine-tuning, which may explain the collapse. This phenomenon is a well-observed problem of DPO (Chen et al., 2024a; Pal et al., 2024), stemming from its focus on optimizing only the *relative* likelihood between preferred and dispreferred data, rather than controlling likelihood for positive and negative image-condition pairs separately. We refer interested readers to Chen et al. (2024a) for a detailed discussion.

**Unlearning.** Also known as unlikelihood training, this method maximizes $\log p_\theta(x|c)$ through standard maximum likelihood training on positive data, while minimizing $\log p_\theta(x|c^{\text{neg}})$ to *unlearn* negative data. A training parameter $\lambda_u$ controls the weight of the unlearning loss. We find that with small $0.01 \leq \lambda_u \leq 0.1$, Unlearning provides some benefit, but it is far less effective than CCA. This suggests the necessity of including a frozen reference model.

## 5.4 Integration of CCA and CFG

If extra sampling costs and design inconsistencies of CFG are not concerns, could CCA still be helpful? A takeaway conclusion is yes: CCA+CFG generally outperforms CFG (Figure 6), but it requires distinct hyperparameter choices compared with CCA-only training.

**Implementation.** After pretraining the unconditional AR visual model by randomly dropping conditions, CFG requires us to also fine-tune the unconditional model during alignment. To achieve this, we follow previous approaches by also randomly replacing data conditions with [MASK] tokens at a probability of 10%. These unconditional samples are treated as positive data during CCA training. We provide pseudo-code in Appendix D.

**Comparison of CCA-only and CCA+CFG.** They require different hyperparameters. As shown in Figure 5, a larger $\lambda$ is typically needed for optimal FID scores in guidance-free generation. For models optimized for guidance-free sampling, adding CFG guidance does not further reduce the FID score. However, with a smaller $\lambda$, CCA+CFG could outperform the CFG method.

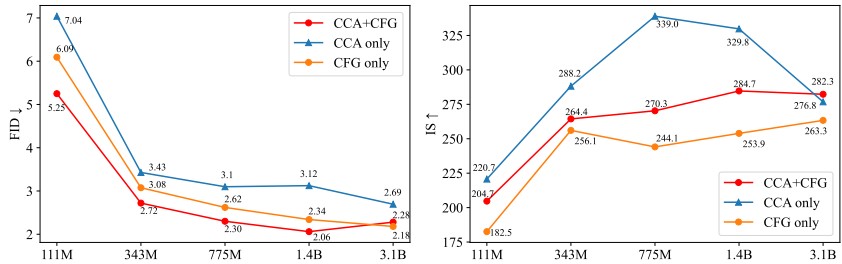

Figure 6: Integration of CCA+CFG yields improved performance over CFG alone.

## 6 RELATED WORKS

**Visual generative models.** Generative adversarial networks (GANs) (Goodfellow et al., 2014; Brock et al., 2018; Karras et al., 2019; Kang et al., 2023) and diffusion models (Ho et al., 2020; Song & Ermon, 2019; Song et al., 2020; Dhariwal & Nichol, 2021; Kingma & Gao, 2024) are representative modeling methods for visual content generation, widely recognized for their ability to produce realistic and artistic images (Sauer et al., 2022; Ho et al., 2022; Ramesh et al., 2022; Podell et al., 2023). However, because these methods are designed for continuous data like images, they struggle to effectively model discrete data such as text, limiting the development of unified multimodal models for both vision and language. Recent approaches seek to address this by integrating diffusion models with language models (Team, 2024; Li et al., 2024; Zhou et al., 2024). Another line of works (Chang et al., 2022; 2023; Yu et al., 2023; Xie et al., 2024; Ramesh et al., 2021; Yu et al., 2022) explores discretizing images (Van Den Oord et al., 2017; Esser et al., 2021) and directly applying language models such as BERT-style (Devlin et al., 2018) masked-prediction models and GPT-style (Radford et al., 2018) autoregressive models for image generation.

**Language model alignment.** LLMs primarily employ training-based alignment techniques to improve instruction-following abilities (Touvron et al., 2023b; OpenAI, 2023). Reinforcement Learning (RL) is well-suited for aligning LLMs with human feedback (Schulman et al., 2017; Ouyang et al., 2022). However, this method requires learning a reward model before optimizing LLMs, leading to an indirect two-stage optimization process. Recent developments in alignment techniques (Rafailov et al., 2023; Chen et al., 2024a; Ji et al., 2024) have streamlined this process. They enable direct alignment of LMs through a singular loss. Among all LLM alignment algorithms, our method is perhaps most similar to NCA (Chen et al., 2024a). Both NCA and CCA are grounded in the NCE framework (Gutmann & Hyvärinen, 2012). Their differences are mainly empirical regarding loss implementations, particularly in how to estimate expectations under the product of two marginal distributions.

**Visual alignment.** Motivated by the success of alignment techniques in LLMs, several studies have also investigated aligning visual generative models with human preferences using RL (Black et al., 2023a; Xu et al., 2024) or DPO (Black et al., 2023b; Wallace et al., 2023). For diffusion models, the application is not straightforward and must rely on some theoretical approximations, as diffusion models do not allow direct likelihood calculation, which is required by most LLM alignment algorithms (Chen et al., 2024b).

## 7 CONCLUSION

In this paper, we propose Condition Contrastive Alignment (CCA) as a fine-tuning algorithm for AR visual generation models. CCA can significantly enhance the guidance-free sample quality of pretrained models without any modification of the sampling process. This paves the way for further development in multimodal generative models and cuts the cost of AR visual generation by half in comparison to CFG. Our research also highlights the strong theoretical connection between language-targeted alignment and visual-targeted guidance methods, facilitating future research of unifying visual modeling and language modeling.

ACKNOWLEDGEMENT

We thank Fan Bao, Kai Jiang, Xiang Li, Min Zhao, Keyu Tian and Kaiwen Zheng for their valuable suggestions and discussion. This work was supported by the National Natural Science Foundation of China (Nos. 62350080, 62106120, 62061136001, 92370124, 92248303, 92270001), BNRist (BNR2022RC01006), Tsinghua Institute for Guo Qiang, and the High Performance Computing Center, Tsinghua University; J. Z was also supported by the XPlorer Prize.

REPRODUCIBILITY

We provide experimental details in Appendix E. We submit our source code in the supplementary material. Code and model weights are publicly accessible.

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

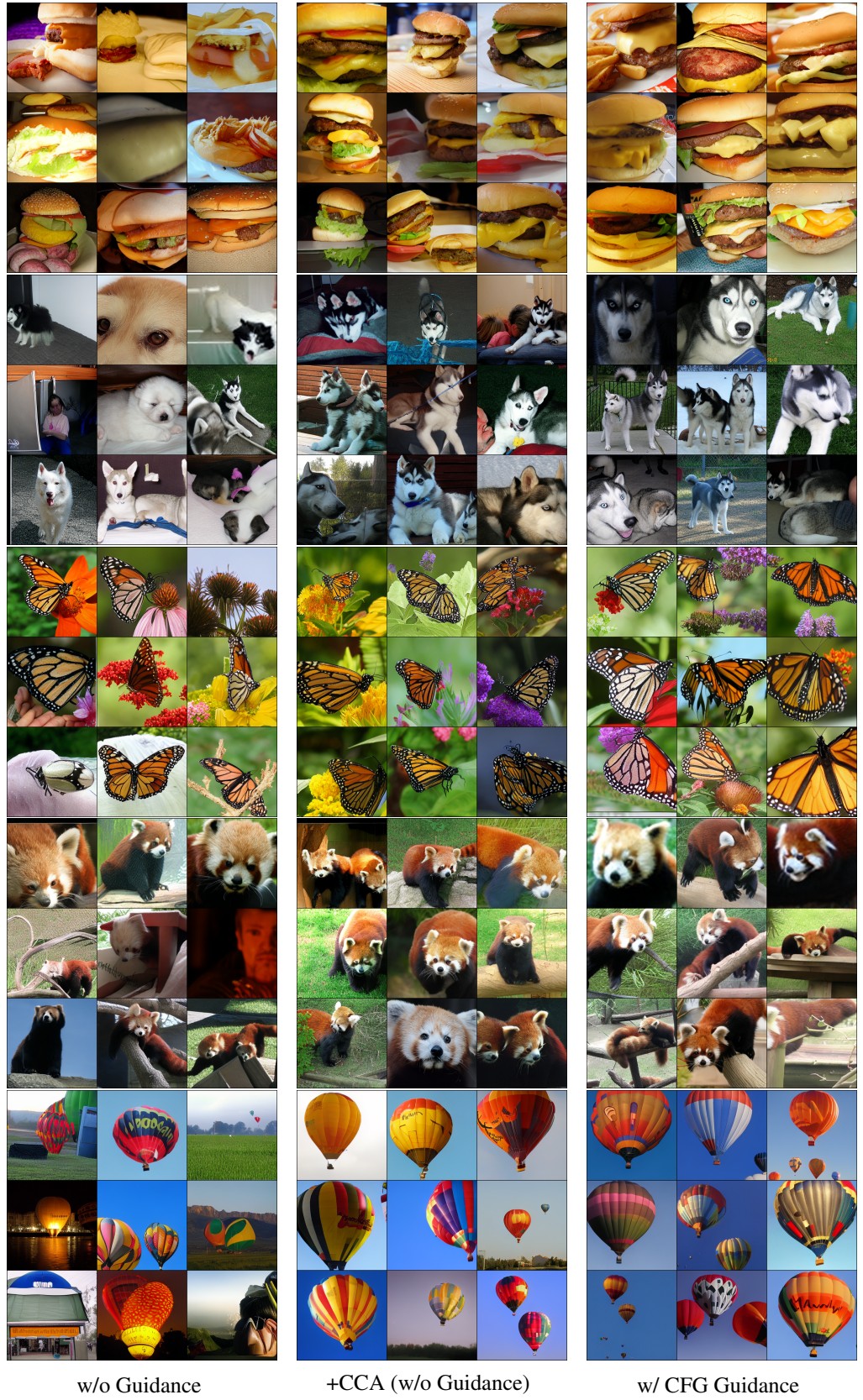

w/o Guidance          +CCA (w/o Guidance)          w/ CFG Guidance

Figure 7: Comparison of LlamaGen-L samples generated with CCA or CFG.

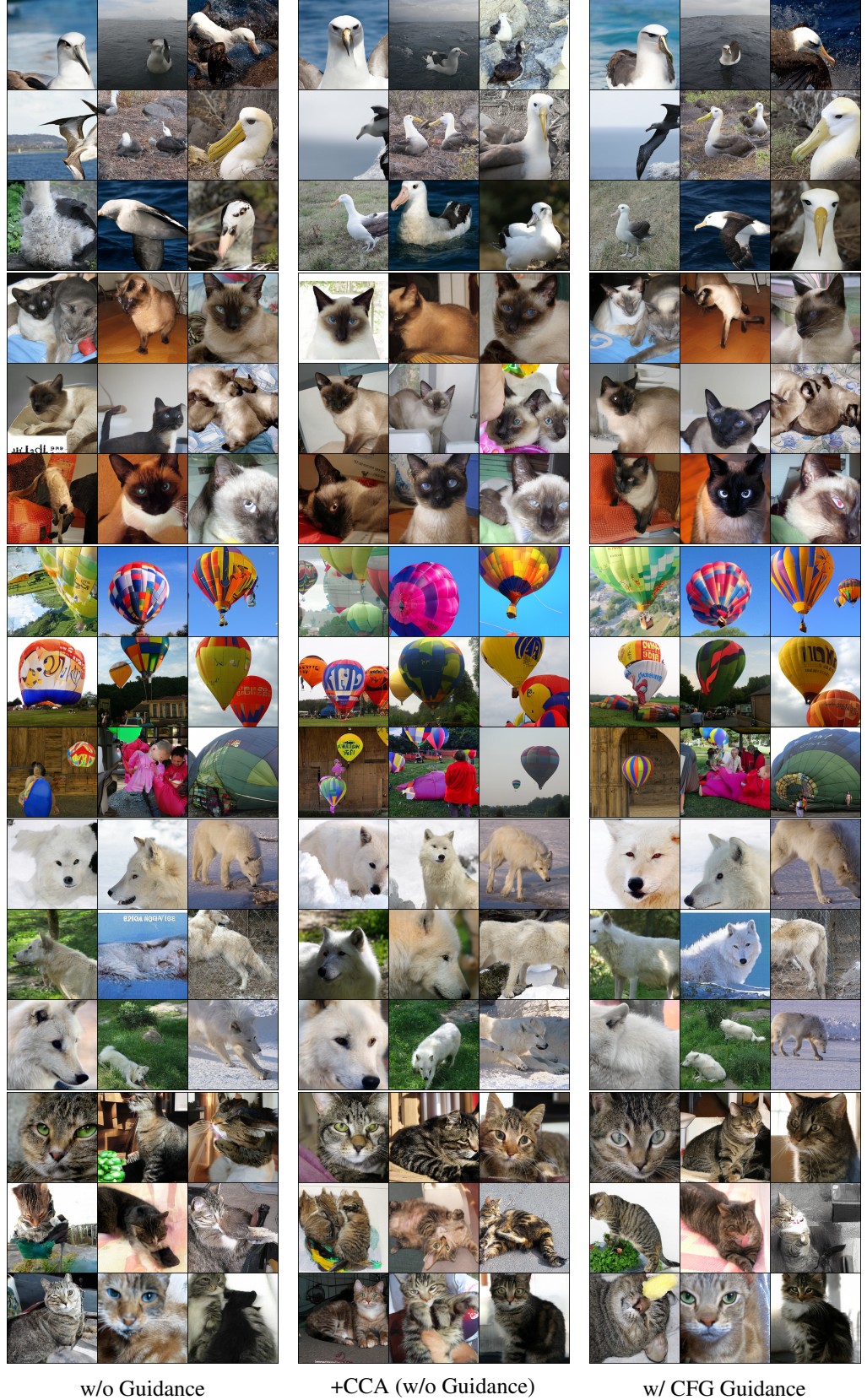

w/o Guidance          +CCA (w/o Guidance)          w/ CFG Guidance

Figure 8: Comparison of VAR-d24 samples generated with CCA or CFG.

# A    THEORETICAL PROOFS

In this section, we provide the proof of Theorem 3.1.

**Theorem A.1** (Noise Contrastive Estimation ). *Let $r_\theta$ be a parameterized model which takes in an image-condition pair $(\boldsymbol{x}, \boldsymbol{c})$ and outputs a scalar value $r_\theta(\boldsymbol{x}, \boldsymbol{c})$. Consider the loss function:*

$$\mathcal{L}_\theta^{NCE}(\boldsymbol{x}, \boldsymbol{c}) = -\mathbb{E}_{p(\boldsymbol{x}, \boldsymbol{c})} \log \sigma(r_\theta(\boldsymbol{x}, \boldsymbol{c})) - \mathbb{E}_{p(\boldsymbol{x})p(\boldsymbol{c})} \log \sigma(-r_\theta(\boldsymbol{x}, \boldsymbol{c})). \tag{13}$$

*Given unlimited model expressivity for $r_\theta$, the optimal solution for minimizing $\mathcal{L}_\theta^{NCE}$ satisfies*

$$r_\theta^*(\boldsymbol{x}, \boldsymbol{c}) = \log \frac{p(\boldsymbol{x}|\boldsymbol{c})}{p(\boldsymbol{x})}. \tag{14}$$

*Proof.* First, we construct two binary (Bernoulli) distributions:

$$Q_{\boldsymbol{x}, \boldsymbol{c}} := \{ \frac{p(\boldsymbol{x}, \boldsymbol{c})}{p(\boldsymbol{x}, \boldsymbol{c}) + p(\boldsymbol{x})p(\boldsymbol{c})}, \frac{p(\boldsymbol{x})p(\boldsymbol{c})}{p(\boldsymbol{x}, \boldsymbol{c}) + p(\boldsymbol{x})p(\boldsymbol{c})} \} = \{ \frac{p(\boldsymbol{x}|\boldsymbol{c})}{p(\boldsymbol{x}|\boldsymbol{c}) + p(\boldsymbol{x})}, \frac{p(\boldsymbol{x})}{p(\boldsymbol{x}|\boldsymbol{c}) + p(\boldsymbol{x})} \}$$

$$P_{\boldsymbol{x}, \boldsymbol{c}}^\theta := \{ \frac{e^{r_\theta(\boldsymbol{x}, \boldsymbol{c})}}{e^{r_\theta(\boldsymbol{x}, \boldsymbol{c})} + 1}, \frac{1}{e^{r_\theta(\boldsymbol{x}, \boldsymbol{c})} + 1} \} = \{ \sigma(r_\theta(\boldsymbol{x}, \boldsymbol{c})), 1 - \sigma(r_\theta(\boldsymbol{x}, \boldsymbol{c})) \}$$

Then we rewrite $\mathcal{L}_\theta^{\text{NCE}}(\boldsymbol{x}, \boldsymbol{c})$ as

$$\mathcal{L}_\theta^{\text{NCE}}(\boldsymbol{x}, \boldsymbol{c}) = -\mathbb{E}_{p(\boldsymbol{x}, \boldsymbol{c})} \log \sigma(r_\theta(\boldsymbol{x}, \boldsymbol{c})) - \mathbb{E}_{p(\boldsymbol{x})p(\boldsymbol{c})} \log \sigma(-r_\theta(\boldsymbol{x}, \boldsymbol{c}))$$

$$= -\int \Big[ p(\boldsymbol{x}, \boldsymbol{c}) \log \sigma(r_\theta(\boldsymbol{x}, \boldsymbol{c})) + p(\boldsymbol{x})p(\boldsymbol{c}) \log \sigma(-r_\theta(\boldsymbol{x}, \boldsymbol{c})) \Big] \mathrm{d}\boldsymbol{x}\mathrm{d}\boldsymbol{c}$$

$$= -\int \Big[ (p(\boldsymbol{x}, \boldsymbol{c}) + p(\boldsymbol{x})p(\boldsymbol{c})) \Big]$$

$$\Big[ \frac{p(\boldsymbol{x}, \boldsymbol{c})}{p(\boldsymbol{x}, \boldsymbol{c}) + p(\boldsymbol{x})p(\boldsymbol{c})} \log \sigma(r_\theta(\boldsymbol{x}, \boldsymbol{c})) + \frac{p(\boldsymbol{x})p(\boldsymbol{c})}{p(\boldsymbol{x}, \boldsymbol{c}) + p(\boldsymbol{x})p(\boldsymbol{c})} \log \big[ 1 - \sigma(r_\theta(\boldsymbol{x}, \boldsymbol{c})) \big] \Big] \mathrm{d}\boldsymbol{x}\mathrm{d}\boldsymbol{c}$$

$$= \int \Big[ (p(\boldsymbol{x}, \boldsymbol{c}) + p(\boldsymbol{x})p(\boldsymbol{c})) \Big] H(Q_{\boldsymbol{x}, \boldsymbol{c}}, P_{\boldsymbol{x}, \boldsymbol{c}}^\theta) \mathrm{d}\boldsymbol{x}\mathrm{d}\boldsymbol{c}$$

$$= \int \Big[ (p(\boldsymbol{x}, \boldsymbol{c}) + p(\boldsymbol{x})p(\boldsymbol{c})) \Big] \Big[ D_{\text{KL}}(Q_{\boldsymbol{x}, \boldsymbol{c}} \| P_{\boldsymbol{x}, \boldsymbol{c}}^\theta) + H(Q_{\boldsymbol{x}, \boldsymbol{c}}) \Big] \mathrm{d}\boldsymbol{x}\mathrm{d}\boldsymbol{c}$$

Here $H(Q_{\boldsymbol{x}, \boldsymbol{c}}, P_{\boldsymbol{x}, \boldsymbol{c}}^\theta)$ represents the cross-entropy between distributions $Q_{\boldsymbol{x}, \boldsymbol{c}}$ and $P_{\boldsymbol{x}, \boldsymbol{c}}^\theta$. $H(Q_{\boldsymbol{x}, \boldsymbol{c}})$ is the entropy of $Q_{\boldsymbol{x}, \boldsymbol{c}}$, which can be regarded as a constant number with respect to parameter $\theta$. Due to the theoretical properties of KL-divergence, we have

$$\mathcal{L}_\theta^{\text{NCE}}(\boldsymbol{x}, \boldsymbol{c}) = \int \Big[ (p(\boldsymbol{x}, \boldsymbol{c}) + p(\boldsymbol{x})p(\boldsymbol{c})) \Big] \Big[ D_{\text{KL}}(Q_{\boldsymbol{x}, \boldsymbol{c}} \| P_{\boldsymbol{x}, \boldsymbol{c}}^\theta) + H(Q_{\boldsymbol{x}, \boldsymbol{c}}) \Big] \mathrm{d}\boldsymbol{x}\mathrm{d}\boldsymbol{c}$$

$$\geq \int \Big[ (p(\boldsymbol{x}, \boldsymbol{c}) + p(\boldsymbol{x})p(\boldsymbol{c})) \Big] H(Q_{\boldsymbol{x}, \boldsymbol{c}}) \mathrm{d}\boldsymbol{x}\mathrm{d}\boldsymbol{c}$$

constantly hold. The equality holds if and only if $Q_{\boldsymbol{x}, \boldsymbol{c}} = P_{\boldsymbol{x}, \boldsymbol{c}}^\theta$, such that

$$\sigma(r_\theta(\boldsymbol{x}, \boldsymbol{c})) = \frac{e^{r_\theta(\boldsymbol{x}, \boldsymbol{c})}}{e^{r_\theta(\boldsymbol{x}, \boldsymbol{c})} + 1} = \frac{p(\boldsymbol{x}, \boldsymbol{c})}{p(\boldsymbol{x}, \boldsymbol{c}) + p(\boldsymbol{x})p(\boldsymbol{c})}$$

$$r_\theta(\boldsymbol{x}, \boldsymbol{c}) = \log \frac{p(\boldsymbol{x}, \boldsymbol{c})}{p(\boldsymbol{x})p(\boldsymbol{c})} = \log \frac{p(\boldsymbol{x}|\boldsymbol{c})}{p(\boldsymbol{x})}$$

$\square$

## B  THEORETICAL ANALYSES OF THE NORMALIZING CONSTANT

We omit a normalizing constant in Eq. 7 for brevity when deriving CCA. Strictly speaking, the target sampling distribution should be:

$$p^{\text{sample}}(\boldsymbol{x}|\boldsymbol{c}) = \frac{1}{Z(\boldsymbol{c})} p(\boldsymbol{x}|\boldsymbol{c}) [\frac{p(\boldsymbol{x}|\boldsymbol{c})}{p(\boldsymbol{x})}]^s,$$

such that

$$\frac{1}{s} \log \frac{p^{\text{sample}}(\boldsymbol{x}|\boldsymbol{c})}{p(\boldsymbol{x}|\boldsymbol{c})} = \log \frac{p(\boldsymbol{x}|\boldsymbol{c})}{p(\boldsymbol{x})} - \frac{1}{s} \log Z(\boldsymbol{c}).$$

The normalizing constant $Z(\boldsymbol{c})$ ensures that $p^{\text{sample}}(\boldsymbol{x}|\boldsymbol{c})$ is properly normalized, i.e., $\int p^{\text{sample}}(\boldsymbol{x}|\boldsymbol{c})\mathrm{d}\boldsymbol{x} = 1$. We have $Z(\boldsymbol{c}) = \int p(\boldsymbol{x}|\boldsymbol{c})[\frac{p(\boldsymbol{x}|\boldsymbol{c})}{p(\boldsymbol{x})}]^s \mathrm{d}\boldsymbol{x} = \mathbb{E}_{p(\boldsymbol{x}|\boldsymbol{c})}[\frac{p(\boldsymbol{x}|\boldsymbol{c})}{p(\boldsymbol{x})}]^s$.

To mitigate the additional effects introduced by $Z(\boldsymbol{c})$, in our practical algorithm, we introduce a new training parameter $\lambda$ to bias the optimal solution for Noise Contrastive Estimation. Below, we present a result that is stronger than Theorem 3.1.

**Theorem B.1.** *Let $\lambda_{\boldsymbol{c}} > 0$ be a scalar function conditioned only on $\boldsymbol{c}$. Consider the loss function:*

$$\mathcal{L}_{\theta}^{NCE}(\boldsymbol{x}, \boldsymbol{c}) = -\mathbb{E}_{p(\boldsymbol{x}, \boldsymbol{c})} \log \sigma(r_{\theta}(\boldsymbol{x}, \boldsymbol{c})) - \lambda_{\boldsymbol{c}} \mathbb{E}_{p(\boldsymbol{x})p(\boldsymbol{c})} \log \sigma(-r_{\theta}(\boldsymbol{x}, \boldsymbol{c})). \tag{15}$$

*Given unlimited model expressivity for $r_{\theta}$, the optimal solution for minimizing $\mathcal{L}_{\theta}^{NCE}$ satisfies*

$$r_{\theta}^*(\boldsymbol{x}, \boldsymbol{c}) = \log \frac{p(\boldsymbol{x}|\boldsymbol{c})}{p(\boldsymbol{x})} - \log \lambda_{\boldsymbol{c}}. \tag{16}$$

*Proof.* We omit the full proof here, as it requires only a redefinition of the distributions $Q_{\boldsymbol{x},\boldsymbol{c}}$ from the proof of Theorem A.1:

$$Q_{\boldsymbol{x},\boldsymbol{c}} := \{\frac{p(\boldsymbol{x}, \boldsymbol{c})}{p(\boldsymbol{x}, \boldsymbol{c}) + \lambda_{\boldsymbol{c}} p(\boldsymbol{x}) p(\boldsymbol{c})}, \frac{\lambda_{\boldsymbol{c}} p(\boldsymbol{x}) p(\boldsymbol{c})}{p(\boldsymbol{x}, \boldsymbol{c}) + \lambda_{\boldsymbol{c}} p(\boldsymbol{x}) p(\boldsymbol{c})}\} = \{\frac{p(\boldsymbol{x}|\boldsymbol{c})}{p(\boldsymbol{x}|\boldsymbol{c}) + \lambda_{\boldsymbol{c}} p(\boldsymbol{x})}, \frac{\lambda_{\boldsymbol{c}} p(\boldsymbol{x})}{p(\boldsymbol{x}|\boldsymbol{c}) + \lambda_{\boldsymbol{c}} p(\boldsymbol{x})}\}$$

Then we can follow the steps in the proof of Theorem A.1 to arrive at the result. □

If let $\lambda_{\boldsymbol{c}} := Z(\boldsymbol{c})^{\frac{1}{s}} = \left[\mathbb{E}_{p(\boldsymbol{x}|\boldsymbol{c})}[\frac{p(\boldsymbol{x}|\boldsymbol{c})}{p(\boldsymbol{x})}]^s\right]^{\frac{1}{s}}$, we could guarantee the convergence of $p_{\theta}^{\text{sample}}$ to $p^{\text{sample}}$. However, in practice estimating $Z(\boldsymbol{c})$ could be intricately difficult, so we formalize $\lambda_{\boldsymbol{c}}$ as a training parameter, resulting in our practical algorithm in Eq. 12.

## C  ADDITIONAL EXPERIMENTAL RESULTS

We provide more image samples to compare CCA and CFG in Figure 7 and Figure 8.

We illustrate the effect of training parameter $\beta$ on the FID-IS trade-off in Figure 9. Overall, $\beta$ affects the fidelity-diversity trade-off similar to $\lambda$ and the CFG method.

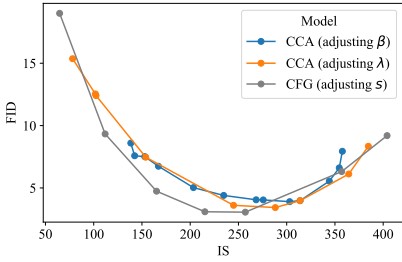

Figure 9: Effect of varying $\beta$ of CCA for the LlamaGen-L model. In our CCA experiments, we either fix $\lambda = 1e3$ and ablate $\beta \in [2, 5e-3]$ (from left to right) or fix $\beta = 0.02$ and ablate $\lambda \in [0, 1e4]$. In our CFG experiments, we ablate $s \in [0, 3]$.

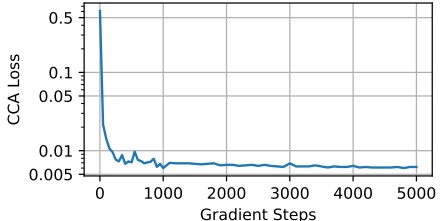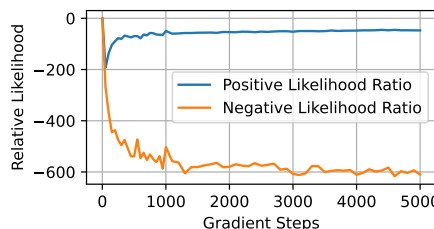

Figure 10: Training curves of CCA for LlamaGen-L model ($\beta = 0.02$, $\lambda = 300$). Left: CCA loss. Right: Relative likelihood $\log \frac{p_\theta(\boldsymbol{x}|\boldsymbol{c})}{p_\phi(\boldsymbol{x}|\boldsymbol{c})}$ for positive and negative data during training.

## D  PSEUDO CODE

---

**Algorithm 1** CCA

---

**Input:** Pretraining dataset $\{\boldsymbol{x}, \boldsymbol{c}\}$, pretrained AR model $p_\phi$, target model $p_\theta$. Initialize $\theta = \phi$
**For** each gradient step **do**
    Sample $K$ data pairs $\{\boldsymbol{x}, \boldsymbol{c}\}_{1:K}$ from the dataset as positive samples // $p(\boldsymbol{x}, \boldsymbol{c})$
    Randomly shuffle $\boldsymbol{c}_{1:K}$ to become $\boldsymbol{c}_{1:K}^{\text{neg}}$ and form negative samples $\{\boldsymbol{x}, \boldsymbol{c}^{\text{neg}}\}_{1:K}$. // $p(\boldsymbol{x})p(\boldsymbol{c})$
    **If** CCA+CFG **then**
        **For** each label $c_k$ in $\boldsymbol{c}_{1:K}$ and $\boldsymbol{c}_{1:K}^{\text{neg}}$ **do**
            Replace $c_k$ with $\varnothing$ with a probability of 10% // Random masking
    $L_\theta = 0$
    **For** For each data $\{\boldsymbol{x}_k, \boldsymbol{c}_k\}$ in training batch $\{\boldsymbol{x}, \boldsymbol{c}\}_{1:K}$ and $\{\boldsymbol{x}, \boldsymbol{c}^{\text{neg}}\}_{1:K}$ **do**
        $L_\theta = L_\theta - \log \sigma \left[ \beta \log \frac{p_\theta^{\text{sample}}(\boldsymbol{x}_k|\boldsymbol{c}_k)}{p_\phi(\boldsymbol{x}_k|\boldsymbol{c}_k)} \right]$ **if** $\{\boldsymbol{x}_k, \boldsymbol{c}_k\}$ is positive sample **or** $\boldsymbol{c}_k = \varnothing$
        $L_\theta = L_\theta - \lambda \log \sigma \left[ - \beta \log \frac{p_\theta^{\text{sample}}(\boldsymbol{x}_k|\boldsymbol{c}_k)}{p_\phi(\boldsymbol{x}_k|\boldsymbol{c}_k)} \right]$ **if** $\{\boldsymbol{x}_k, \boldsymbol{c}_k\}$ is negative sample **and** $\boldsymbol{c}_k \neq \varnothing$
    $\theta \leftarrow \theta - \eta \nabla_\theta L_\theta$ (Eq. 12)

---

We provide an example of training curves for CCA in Figure 10.

## E  TRAINING HYPERPARAMETERS

Table 4 reports hyperparameters for chosen models in Figure 1 and Figure 6. Other unmentioned design choices and hyperparameters are consistent with the default setting for LlamaGen `https://github.com/FoundationVision/LlamaGen` and VAR `https://github.com/FoundationVision/VAR` repo. All models are fine-tuned for 1 epoch on the ImageNet dataset. We use a mix of NVIDIA-H100, NVIDIA A100, and NVIDIA A40 GPU cards for training.

| Type | LlamaGen | | | | | VAR | | | |
|---|---|---|---|---|---|---|---|---|---|
| Model | B | L | XL | XXL | 3B | d16 | d20 | d24 | d30 |
| Size | 111M | 343M | 775M | 1.4B | 3.1B | 310M | 600M | 1.0B | 2.0B |
| CCA $\beta$ | 0.02 | 0.02 | 0.02 | 0.02 | 0.02 | 0.02 | 0.02 | 0.02 | 0.02 |
| CCA $\lambda$ | 1000 | 300 | 1000 | 1000 | 500 | 50 | 50 | 100 | 1000 |
| CCA+CFG $\beta$ | 0.1 | 0.02 | 0.1 | 0.1 | 0.1 | - | - | - | - |
| CCA+CFG $\lambda$ | 1 | 1 | 1 | 1 | 1 | - | - | - | - |
| Learning rate | 1e-5 | 1e-5 | 1e-5 | 1e-5 | 1e-5 | 2e-5 | 2e-5 | 2e-5 | 2e-5 |
| Dropout? | Yes | Yes | Yes | Yes | Yes | None | Yes | Yes | Yes |
| Batch size | 256 | 256 | 256 | 256 | 256 | 256 | 256 | 256 | 256 |

Table 4: Hyperparameter table.

All our reported models are trained individually for each hyperparameter. However, we note that hyperparameters like $\lambda$ and $\beta$ can serve as input for our target AR visual model using existing

distillation techniques (Meng et al., 2023) so that we can tune them only during inference. This way CCA can allow test-time flexibility just like CFG. We present an initial result conditioning the LlamaGen-L model on parameter $\lambda$ in Table 5. In order to additionally condition on an extra scalar input $\lambda$, we use the same embedding method as the one used by DiT (Peebles & Xie, 2023) and directly add the $\lambda$ embedding on the class token embeddings. We randomly sample $\lambda \in [e^0, e^9 \approx 10000]$ during training. The model is trained for 3 epochs.

| Inference-time $\lambda$ | 10 | 100 | 300 (Chosen) | 1000 | 3000 | 10000 |
|---|---|---|---|---|---|---|
| FID | 7.23 | 4.18 | **3.59** | 3.73 | 4.12 | 5.33 |
| IS | 153.1 | 218.8 | **256.2** | 277.5 | 307.7 | 341.2 |

Table 5: Performance for different inference-time $\lambda$ values. For reference, the pretrained LlamaGen model has an IS of 64.3 and an FID of 19.07. After CCA finetuning with fixed $\lambda = 300$, the finetuned model has an IS 288.2 of and FID of 3.43.

