# OpenReview forum: "Toward Guidance-Free AR Visual Generation via Condition Contrastive Alignment"
_ICLR.cc/2025/Conference — ICLR 2025 Oral_

### Official Review · Reviewer_qqAa · 2024-10-24

**Soundness:** 3
**Presentation:** 3
**Contribution:** 3
**Rating:** 8
**Confidence:** 3

**Summary:**

This submission proposes an alternative way to achieve classifier-free guidance (CFG) for visual autoregressive model, by directly finetuning the model, which avoids an extra forward pass when deploying CFG. The experiment results show the proposed finetuning method introduces huge performance gain.

**Strengths:**

- The problem the submission addresses has very high practical value to deploying the visual autoregressive model.
- The algorithm derivation gives a clear explanation about the methods. And the derived training loss function is very concise, which is easy to train.
- The performance gain is huge, on both generation quality and diversity.
- Experimental section is thorough. Additionally, sec 5.3 is informative and appreciated, especially unlearning.

**Weaknesses:**

In short, this is a good work. I only have one question about the training loss function, as $\beta$ is expected to be responsible for CFG degree. But the experimental results show that $\lambda$ actually dominates the CFG controllability, any clue about it?

One open question is, will the proposed method also improve the directional autoregressive model, a.k.a., Masked Generative Models such as MaskGIT and MAGE. Or, to think even more boldly, will the method still applicable to (continuous) diffusion models, since MaskGIT could be regarded as a discrete diffusion model.

**Questions:**

Please check the weakness.

---

> ### Author Response · Authors · 2024-11-18
> **Official Response to Reviewer qqAa**
>
> We thank the reviewer for the valuable comments and seek to address reviewer's concerns below:
>
> **Q1: Why experiments show that $\lambda$ dominates the CFG controllability while $\beta$ is expected to be responsible for that according to the derivations?**
>
> **A1:** We really appreciate the reviewer for bringing up such insightful and meaningful question.
>
> Frankly speaking, we are not entirely sure, but have some guesses:
>
> The loss derivation in our main paper has overlooked the normalization constant of the target distribution, making the $\lambda$ parameter seemingly unnecessary. However, in a more strict theoretical derivation presented in **Appendix B** of our paper, we show that $\lambda$ is actually required and calculating $\lambda^*$ analytically is difficult, so we can only set it as a hyperparameter. Also in our derivations, we show $\lambda^*$ is also correlated with $\beta$. Finding $\lambda$'s best value may affect the training behavior more than tuning $\beta$ itself.
>
> Overall, we are not certain about this conclusion. This is really worth further in-depth study.
>
> **Q2: Would CCA be potentially be applied to Mask AR model like MaskGIT/MAGE?  What about continuous diffusion models?**
>
> **A2:** I believe CCA can be directly applied to Mask AR models, and this is really worth trying out in future work.
>
> Mask AR randomly masks some tokens and predict these masked tokens instead of always predicting next-tokens as in standard AR. Correspondingly, for Mask ARs we should also randomly mask some tokens and apply CCA  loss on these masked tokens. It is basically the same pipeline as standard CCA methods used in our paper.
>
> For continuous diffusion models, removing CFG by fintuning is **exactly** what we are working on right now! However, we do not think CCA can be applied to continuous diffusion models due to its requirement to exactly calculate model likelihood. For diffusion models, we can follow the principle idea of CCA, but need a new method. We expect to share our latest progress in a few months. Our initial results is quite promising.

---

> > ### Comment · Reviewer_qqAa · 2024-11-21
> > **Thanks for the rebuttal**
> >
> > Thank you for the rebuttal, I would be interested in following the subsequent researches, and I keep my score.

---

### Official Review · Reviewer_Zf6F · 2024-10-29

**Soundness:** 3
**Presentation:** 4
**Contribution:** 4
**Rating:** 8
**Confidence:** 4

**Summary:**

The authors propose a novel method for fine-tuning auto-regressive models to enable guidance-free sampling from the same distribution that would be sampled if classifier-free guidance were used. The training objective is derived by noise contrastive estimation in which genuine pairs of condition and data are used as positive while random pairs are negative ones. The experimental results show that just one epoch fine-tuning with the proposed method greatly improves the performance of the guidance-free sampling.

**Strengths:**

- The motivation of this study is clear and should be important to make auto-regressive models efficiently generate high-quality visual contents. As of 2024, CFG is commonly used for auto-regressive models to achieve high-quality image generation, while it doubles the computational cost of the generation process. How to get rid of the necessity of CFG would be an important problem especially for practical applications.

- The derivation of the proposed method, particularly Eq. (10) and (11), is elegant. The theoretical connection with guidance-based sampling is clear, and the practical implementation is reasonable.

- The experimental results validate the advantage of the proposed method; it greatly boosts the performance of guidance-free sampling. Although its performance is slightly worse than that of sampling with CFG, it would be acceptable in some practical applications considering that the computational cost for the generation is halved.

- The manuscript is well-written and easy to follow.

**Weaknesses:**

- I do not find any crucial weakness or major concerns on this paper. The followings are just minor points.
  - As the proposed method requires to fix a target distribution represented by beta, it cannot control the fidelity-diversity trade-off in the inference phase, which might reduce some flexibility compared with CFG.
    - It might be worth exploring whether the model could be extended to accept additional condition beta to work with various beta in the inference phase, similar to the approach in the distillation of guided diffusion models [R1]. However, it should be non-trivial how to achieve such extention in AR models.

      [R1] "On Distillation of Guided Diffusion Models," CVPR 2023.
  - It would be beneficial if the author could discuss how stable the proposed method is. As the negative pairs in Eq. (12) are random pairs of x and c, p(x|c) appeared in the loss computation could have very small values, which might cause numerical instability during the fine-tuning.

**Questions:**

- It is not clearly descirbed why the performance with varying lambda leads to a similar behaviour to that with varying beta. For example, why has IS score been improved by increasing the value of lambda(namely, overweighing the loss on negative pairs)?

---

> ### Author Response · Authors · 2024-11-18
> **Official Response to Reviewer Zf6F**
>
> We thank the reviewer for the valuable comments and seek to address reviewer's concerns below:
>
> **Q1: It might be worth exploring whether the model could be extended to accept additional condition beta to increase flexibility of trade-off during inference.  Similar to On Distillation of Guided Diffusion Models," CVPR 2023.**
>
> **A1:** We thank the reviewer for this very helpful and insightful comment!  The CVPR paper inspires us a lot.
> It is indeed not easy to do so similarly on AR. For diffusion, $\lambda$ can be injected similarly like time $t$. To solve this, we experimented naively adding the $\lambda$ information to conditional embedding tokens, after $\lambda$ is transformed by some GaussianFourier Prejection followed by some layers of MLPs.
>
> Below is our (very initial) result:
>
> New experimental result: Flexible $\lambda$ as input, (only 1 model, 3 epochs training):
>
> | Inference-time $\lambda$  |   Base Model    | 10     | 100   | 300 (Chosen) | 1000  | 3000  | 10000 |
> |------------|-------------|--------|-------|--------------|-------|-------|-------|
> | FID        | 19.07       | 7.23   | 4.18  | **3.59**         | 3.73  | 4.12  | 5.33  |
> | IS         | 64.3        | 153.1  | 218.8 | **256.2**        | 277.5 | 307.7 | 341.2 |
>
>
>
> **Q2: How stable the proposed method is. $p(x|c)$ could be a very small value if x and c are randomly drawn from data. Would this lead to numerical instability?**
>
> **A2:** CCA is quite stable in all our experiments. I understand the reviewer's concern is $p(x|c)$ may be too small (e.g., 1e-30) if (x, c) is negative data. We think this is actually not a concern from two observations:
>
> 1. $p(x|c^{\text{neg}})$ is actually comparable to $p(x|c)$ in practice, (though admittedly *slightly* smaller), which **means $p(x|c^{\text{neg}})$ is NOT a small value even $x$ and $c^\text{neg}$ are not correctly paired.** We believe this is due to the model's limited expressivity or certain training limitations. The *pretrained* AR model is not as strong as we expected in distinguishing different labels $c$. We think **this is exactly why, by explicitly "unlearning" negative data likelihood, CCA can achieve a much better performance** than pretrained models.
> 2. CCA loss only care about **relative** likelihood ratio $\log \frac{p_\theta}{p_\phi}$ between pretrained and optimized model.
> $Loss =  - \log \sigma[ \beta \log \frac{p_\theta}{p_\phi} (x|c)]$ for positive data.
> $Loss =  - \lambda \log \sigma[ - \beta \log \frac{p_\theta}{p_\phi} (x|c)]$ for negative data.
> If $p_\phi$ is small, $p_\theta$ is also likely to be small. We observe usually $-500<\log \frac{p_\theta}{p_\phi}<200$ for all data.
>
>
> **Q3: Why varying lambda leads to a similar behavior to that with varying beta? Why has IS score been improved by increasing the value of lambda?**
>
> **A3:** Frankly speaking, we are not entirely sure. We have some guesses:
>
> The loss derivation in our main paper has overlooked the normalization constant of the target distribution, making the $\lambda$ parameter seemingly unnecessary. However, in a more strict theoretical derivation presented in **Appendix B** of our paper, we show that $\lambda$ is actually required and calculating $\lambda^*$ analytically is difficult, so we can only set it as a hyperparameter. Also in our derivations, we show $\lambda^*$ is also correlated with $\beta$. Finding $\lambda$'s best value may affect the training behavior more than tuning $\beta$ itself.
>
> Overall, we are not certain about this conclusion. This is really worth further in-depth study.

---

> > ### Comment · Reviewer_Zf6F · 2024-11-19
> > **Thanks for your response**
> >
> > A1: The results seem great. I am glad to see that my comments have helped in extending the proposed model, which also addresses some of the concerns regarding computational cost raised by other reviewers. I believe these results would be worth showing in the appendix or somewhere.
> >
> > A2: I see. It would be beneficial to show some empirical evidence on this (e.g., training loss curves, or actual log-density values observed in the experiments as described in the response) somewhere in the paper/appendix.
> >
> > A3: Thanks for providing some intuition. I understand that the optimal lambda depends on a given beta, and finding it has empirically larger impact on the performance than tuning beta itself. Although it still does not directly explain why IS is monotonically improved as lambda increases, the results are consistent across the two models and empirically support the authors' argument.

---

> > > ### Author Response · Authors · 2024-11-20
> > > **Thanks for your detailed suggestions**
> > >
> > > Thanks for your prompt feedback and detailed suggestions.
> > >
> > > **A1**: Based on your suggestions, we have included the initial flexible $\lambda$ results in Appendix E of our paper (Table5).
> > >
> > > **A2**: Based on your suggestions, we have added the CCA loss curve and relative likelihood curve in Figure 10 of our paper.
> > >
> > > **A3**: We are glad our responses help.

---

### Official Review · Reviewer_CPMH · 2024-11-02

**Soundness:** 3
**Presentation:** 3
**Contribution:** 3
**Rating:** 6
**Confidence:** 4

**Summary:**

The authors propose condition contrastive alignment (CCA), replacing classifier-free guidance (CFG). Based on CCA, they implement a guidance-free AR visual generation and unifies the alignment mechanism between language and visual models. Compared with models using CFG, generation models with CCA can reduce the sampling cost by half. To implement CCA, the authors transform CFG into a contrastive learning formulation and train the sampling model only with a fine-tuning loss. In addition, they conduct extensive experiments to validate the effectiveness of the proposed method.

**Strengths:**

+ This paper proposes condition contrastive alignment (CCA) for a guidance-free AR visual generation. Concretely, the authors transform CFG into a contrastive learning formulation and train the sampling model only with a fine-tuning loss. Thus, CCA is simple and easy to implement.
+ The proposed method can unifies the alignment mechanism between language and visual models.  Besides, unlike CFG, generation models with CCA can reduce the sampling cost by half, which is a practical technique.
+ The authors conduct extensive experiments to validate the proposed method. Compared with CFG, CCA achieves comparable performance in FID score under different visual models.

**Weaknesses:**

- Training cost. CCA needs one training epoch to get ideal performance. The training cost is negligible for ImageNet datasets. But for existing large visual generation models (like SD3), the training data is huge. The cost of training one epoch with large-scale datasets is expensive. Is there a practical solution to reduce the cost?
- Optimal $\beta$ and $\lambda$. To obtain the optimal values for $\beta$ and $\lambda$, we need to train CCA many times. The cost of this process is large. Is there a potential guideline to mitigate the substantial cost of finding optimal $\beta$ and $\lambda$?
- Negative samples in CCA. To my knowledge, negative samples have a large effect on the performance of contrastive learning. For CCA, it is based on contrastive learning framework. The selection of negative samples affects the performance of CCA largely. Is there a selection mechanisms of  negative samples?
- Implementation about integration of CCA and CFG. I do not understand the implementation completely due to lacks of details. It is necessary to clarify the implementation with more details.
- IS score. As shown in Figure 1 and Figure 6, IS score starts to degrade when model size is beyond a certain size, such as LlamaGen>775M. To my knowledge, this is an unusual phenomenon. Larger model means better IS score.

**Questions:**

To make the proposed method better, the authors need to provide practical solutions to some issues like training cost with large-scale datasets.

---

> ### Author Response · Authors · 2024-11-18
> **Official Response to Reviewer CPMH (Part 1/2)**
>
> We thank the reviewer for the valuable comments and seek to address reviewer's concerns below:
>
> **Q1: One epoch training is negligible for ImageNet datasets. But for existing large visual generation models (like SD3), the training data is huge. Is there a practical solution to reduce the cost?**
>
> **A1:** It is not that CCA requires at least one epoch of finetuning in order to be effective. It's just that we find one epoch already sufficient for CCA, which shows CCA only requires 1% of pretraining computation (300 epochs).
>
> **For large datasets like LAION-5B, we believe CCA does not need one whole epoch.** Previous experiences [1,2] in finetuning large AR models on large datasets with ControlNet show that finetuning 5-10k gradient steps is sufficient for good performance.
>
> We understand it's best to give some experimental evidence on LAION-5B dataset here. However, there currently lacks a publicly accessible pretrained foundational t2i visual AR model on public datasets like LAION-5B and we lack enough computing resources in limited rebuttal time. We thus hope the reviewer finds it acceptable that we settled for evaluating the performance of CCA  with <1 epoch training on ImageNet:
>
> | Training Steps | 0 | 1k steps |2k steps|3k steps|4k steps (converges) | 5k steps (1 epoch) | 15 k steps|
> |-------|------|-------------|-----|----------|-----|----------|-----|
> | FID            | 19.1 | 4.71| 3.96 | 3.83|  3.54         | 3.51 | 3.49|
>
> The FID quickly goes down and converges even when the first epoch is not finished. 4K gradient steps is about 1 epoch for ImageNet, but for **LAION-5B it is only less than 0.01 epoch**.
>
> [1]Emu3: Next-Token Prediction is All You Need  https://arxiv.org/abs/2409.18869
>
> [2]ControlVAR: Exploring Controllable Visual Autoregressive Modeling https://arxiv.org/pdf/2406.09750
>
>
> **Q2: We need to train CCA many times to  obtain optimal $\beta$ and $\lambda$. Is there a potential guideline to mitigate the substantial cost of finding optimal $\beta$ and $\lambda$?**
>
> **A2:** We really appreciate the reviewer for bringing up this question:
>
> 1. We find $\beta$ and $\lambda$ affect CCA performance similarly (Appendix C in paper). **It's often not necessary to tune them both if not for academic research**. We recommend fixing $\beta$ and adjusting $\lambda$ only.
> 2. We find $\beta=0.02$ is usually a good choice for all models we test. In fact, ALL reported numbers in our paper use $\beta=0.02$ by default except for ablation studies.
> 3. $\lambda$ has a very stable effect on CCA performance. Starting with [1.0, 10, 100, 100] and using the bisection method to find the optimal value is generally sufficient.
> 4. **CCA is extremely efficient.** Even if we tested 10 points to get a final satisfying model. That is **only 10% of computation/time compared with pretraining.**
>
> **What's more exciting is:** We can make $\lambda$ as input of our model, and randomly sample $\lambda \in [e^0, e^{8}\approx3000]$ during training. This way we can train models with multiple $\lambda$ simultaneously and only test out optimal hyperparameters during inference, **directly solving the above problem.** A similar method has been used by [3].
>
> New experimental result: Flexible $\lambda$ as input, (only 1 model, 3 epochs training):
>
> | Inference-time $\lambda$  |   Base Model    | 10     | 100   | 300 (Chosen) | 1000  | 3000  | 10000 |
> |------------|-------------|--------|-------|--------------|-------|-------|-------|
> | FID        | 19.07       | 7.23   | 4.18  | **3.59**         | 3.73  | 4.12  | 5.33  |
> | IS         | 64.3        | 153.1  | 218.8 | **256.2**        | 277.5 | 307.7 | 341.2 |
>
> Please note that the above results are very initial and without any tuning/ablation.
>
> [3]  On Distillation of Guided Diffusion Models. Chenlin Meng, Robin Rombach, Ruiqi Gao, Diederik Kingma, Stefano Ermon, Jonathan Ho, Tim Salimans. CVPR 2023.

---

> ### Author Response · Authors · 2024-11-18
> **Official Response to Reviewer CPMH (Part 2/2)**
>
> **Q3: Is there a selection mechanism of negative samples? The selection of negative samples may affect the performance of CCA (contrastive learning) largely.**
>
> **A3:** **Not at all!** This is exactly the fascinating part of the CCA algorithm! There aren't **any** special tricks/strategies for constructing negative data.  The negative labels come **solely** from ImageNet by randomly shuffling labels within the same training batch.
>
> Specifically:
> 1. We sample 256 images $[x_i]$ and their labels $[c_i]$ from ImageNet.
> 2. Then we randomly shuffle $[c_i]$ to form negative labels $[c_i^{\text{neg}}]$.
> 3. $[x_i, c_i]$ are original positive data in ImageNet. $[x_i, c_i^{\text{neg}}]$ are new negative data.
>
> Sampling code implementation (only 1 line): `Supplementary\iclr2025submitted\LlamaGen\autoregressive\train\finetune_c2i.py` Line 275
>
> **Q4: I do not understand the implementation of the 'CCA+CFG' algorithm completely due to lack of details. Require more clarifications.**
>
> **A4:** We have added pseudo code to facilitate understanding of our method in Appendix D and thank the review for raising this issue~
>
> The core part of 'CCA+CFG' is that it additionally takes care of training unconditional models compared with 'CCA only':
> 1. We sample 256 images $[x_i]$ and their labels $[c_i]$ from ImageNet.  Then we randomly shuffle $[c_i]$ to form negative labels $[c_i^{\text{neg}}]$.
> 2. We concat $[x_i, x_i]$ and $[c_i, c_i^{\text{neg}}]$. This results in a training batch of 512 including both positive and negative data.
> 3. **If CCA+CFG training:**  We randomly select 10% labels from $[c_i, c_i^{\text{neg}}]$ and replace them with unconditional [MASK] tokens.
> 4. According to Eq 12 in the paper. For each data within 512 training batch: $Loss =  - \log \sigma[ \beta \log \frac{p_\theta}{p_\phi} (x|c)]$ if $c$ is the positive label for $x$, **or if $c$ is unconditional [MASK]**.  Otherwise, we have $Loss =  - \lambda \log \sigma[ - \beta \log \frac{p_\theta}{p_\phi} (x|c)]$ if $c$ is negative label.
>
> The above method ensures for CFG+CCA method. Unconditional models are also trained with a 10% probability. The loss for unconditional data is the same as the loss for positive data. (Basically, you can consider it as Maximum likelihood (MLE) Loss because it always increases likelihood $p_\theta(x|c=[MASK])$. )
>
>
> **Q5: IS score starts to degrade when model size is beyond a certain size in Figure 1. Unusual. Why?**
>
> **A5:** Frankly speaking, we do not entirely know why. We think/guess this is mainly related to hyperparameter choices. We all know that there is a trade-off between IS and FID. In Figure 1, all models are selected to have minimal FID. The IS is not a main focus in hyperparameter selection. For example for LlamaGen-3B model, we can increase IS score from 260 to 300+ by adjusting $\lambda$ parameter and sacrificing some FID score.

---

> > ### Comment · Reviewer_CPMH · 2024-11-21
> > **Thanks for your response.**
> >
> > Thanks authors for the responses. Most of my concerns have been solved. I still have some concerns as follows:
> >
> > A1. It is expected for CCA to obtain acceptable results with less than one epoch of training on ImageNet. Since the concepts (categories) in ImageNet is not large (1000), it is efficient to learn these concept quickly. But for large-scale datasets, especially ones with rich text annotations, it is hard to learn all concepts and relationships with a short training time. The cost of training is a potential limit.
> >
> > A5: Sure, it is well known that there is a trade-off between IS and FID. But to my knowledge, with model size increases, these two metric scores should become better at the same time.

---

> ### Author Response · Authors · 2024-11-21
> **Further reponses**
>
> **Q1: For large-scale datasets, especially ones with rich text annotations, it is hard to learn all concepts and relationships with a short training time.?**
>
> We have some observations which we think may give the reviewer a different perspective.
> 1. **CCA should generalize very well**. I mean, we agree ImageNet only has 1000 classes, but T2I has basically infinite kinds of text annotations, and its concepts are difficult to learn. But consider this, for imagenet, there are a total of 1000\*1000 different combinations for images and labels, and CCA has only met 1000\*2 possibilities in these combinations. Only 0.2%! This shows that CCA generalizes very well on the other 99.8% unseen data space, and we don't know if this is its upper limit yet.
> 2. **Maybe**, T2I generation is not that much harder than C2I generation. I cannot give a very specific reference here. However. we have some experience in finetuning Sora-type text2video models, which should presumably be much more difficult than T2I models. Our observations are that, regarding **fintuning**, 10,000 steps is about enough even for video model to understand a new condition input format (such as an additional starting-frame image). We really don't need to go through all video data used for pretraining, and we don't think T2I models should require this, either.
> 3. Alignment algorithms may generalize much better than we think. CCA is basically an alignment algorithm. There is plenty of evidence in NLP community. Typically for LLMs like Llama2-130B, the pretraining data size/computational cost is HUGE. However, the preference dataset (like 200K pairs) and computational cost (like 8 A100) used for finetuning is much much much smaller. The preference dataset surely cannot cover all the language data space, but LLMs generalize just fine.
>
> Still, we have to say this is really like some kind of "belief" of ours since we indeed do not have direct experimental evidence due to clear difficulties discussed previously.
>
> **Q2: To my knowledge, with model size increases, these two metric scores should become better at the same time.**
>
> Yes, we totally get the reviewer's point. Still, we are not entirely sure about why this happens. We would certainly expect CCA models to outperform CFG in every way, but sadly the experiments show otherwise and we have to faithfully report the results. Maybe this is because of insufficient tunning (We keep most parameters consistent with pretraining and across difference-size models and did not tune them)? Or maybe this is because of problems of pretrained models (If you look closely at LlamaGen base model performance, the IS stopped increasing after >343 M parameters, and for larger models, LlamaGen tune guidance scale $s$ more carefully by using a smaller searching interval.)?
>
> Anyway, we want to stress that CCA is still quite an explorative work in the current stage, we believe **its main contribution is pointing a promising possible direction for future visual alignment works**. After all, we successfully **lowered the FID from 9.38 to 2.69 with one simple loss function** for a 3B model. This is 93% closer to the CFG performance. There are indeed some unsatisfying features of CCA in the current stage. We are actively working on solving related problems.  We would be grateful if the reviewer finds this current performance acceptable and indicative.
>
> ***
> Lastly, we'd also be grateful the reviewer could consider re-evaluating our work if our additional input makes some point.

---

> > ### Comment · Reviewer_CPMH · 2024-11-22
> > **Thanks for your further explanation**
> >
> > Thanks authors for the responses. We can leave A1 and A5 to future work. I prefer to keep my score as 6.

---

> > > ### Author Response · Authors · 2024-11-22
> > >
> > > Thank you for your feedback. We will continue to improve our work.

---

### Official Review · Reviewer_8sAn · 2024-11-03

**Soundness:** 4
**Presentation:** 3
**Contribution:** 3
**Rating:** 6
**Confidence:** 4

**Summary:**

This paper introduces CCA which a new alternative to CFG to finetune autoregressive image generation models so that they work like the sampling model used by CFG during inference. The key idea lies in modeling the conditional residual by learning to maximize the likelihood of an image given positive conditions, while minimizing the likelihood given negative conditions. In my opinion, it can be intuitively seen as a way to absorbing the benefits of the unconditional model (which complements the conditional model as demonstrated by CFG) right into single conditional model inference. The resulting approach is more elegant than CFG and is more efficient than CFG because it only requires 1x sampling cost. Furthermore, it can be paired with CFG for additional inference quality boost.

**Strengths:**

- The paper tackles an important but often overlooked problem in autoregressive image generation.

- The methodology section of the paper is mostly well written except that it lacks key information in certain areas. Overall, it provides strong motivations and reasonable derivations based on prior work that lead to the proposed CCA method.

- The experimental results using strong AR visual generation models (LLamaGen, VAT) as baselines on ImageNet are very promising. It is impressive that without any additional inference costs (and a bit more training/finetuning costs), CCA can bring significant improvements to the FID and IS metrics. When paired with CFG, improved FIDs can be achieved.

**Weaknesses:**

- Some important details are missing in the paper, such as how positive and negative data samples are chosen for CCA training. Sec 3.2 only briefly mentions that and Fig. 2 shows something about it without any verbal explanation. It is unclear if we just use the ImageNet labels as the sole criterion for constructing positives and negatives or there’s some other strategy being used. In Sec 5.4, it’s mentioned that certain unconditional data samples are used as positives but it is not clear how exactly that works. Without this key information, it may be hard for readers to understand how and why CCA works.

- Fig 2‘s caption is totally uninformative. There should be a caption describing the method in detail.

- CCA uses models pretrained with CFG training (i.e., condition dropout). For CCA to work, is there a strong requirement on this? What is the relationship of CFG pretraining and CCA finetuning?

- When CCA is combined with CFG, IS becomes worse while FID becomes better. Can the authors explain this phenomenon?

- The proposed method resembles CLIP-like contrastive loss. I wonder if it can only work if the training batch size is large enough.

**Questions:**

Please refer to weaknesses

---

> ### Author Response · Authors · 2024-11-18
> **Official Response to Reviewer 8sAn (Part 1/2)**
>
> We thank the reviewer for the valuable comments and seek to address reviewer's concerns below:
>
> **Q1: Details are missing: how positive and negative data samples are chosen for CCA training? Do we just use the ImageNet labels as the sole criterion for constructing positives and negatives or there’s some other strategy being used? "**
>
> **A1:**
> There aren't **any** special tricks/strategies for constructing negative data. This is exactly the fascinating part of the CCA algorithm! The negative labels come **solely** from ImageNet by randomly shuffling labels within the same training batch.  Data sampling method is **exactly as described in Line 240-244 in the paper**. We strictly follow the derived loss function when implementing CCA.
>
> We have added pseudo code in Appendix D of our paper to facilitate understanding of our method and thank the review for raising this issue~
>
> Simple version:
> 1. Sample 256 images $[x_i]$ and their labels $[c_i]$ from ImageNet.  Then we **randomly shuffle** $[c_i]$ to form negative labels $[c_i^{\text{neg}}]$.
> 2. We concat $x = [x_i, x_i]$ and $c=[c_i, c_i^{\text{neg}}]$. This results in a training batch $[x,c]$ of 512 data including both positive and negative data.
> 3. According to Eq 12 in the paper. For each data within 512 training batch: $Loss =  - \log \sigma[ \beta \log \frac{p_\theta}{p_\phi} (x|c)]$ if $c$ is the positive label for $x$.  Otherwise, we have $Loss =  - \lambda \log \sigma[ - \beta \log \frac{p_\theta}{p_\phi} (x|c)]$ if $c$ is negative label.
>
> Sampling Python code (only 1 line): `Supplementary\LlamaGen\autoregressive\train\finetune_c2i.py` Line 275
>
> **Q2: CCA uses models pretrained with CFG training (i.e., condition dropout). For CCA to work, is there a strong requirement on this? What is the relationship of CFG pretraining and CCA finetuning?"**
>
> **A2:**
> **This is NO requirement for CFG training in order to apply CCA (only).**
>
> **For 'CCA only' experiments:**
> Ideal pipeline is:    Normal pretraining **wo/** CFG  **+**  CCA finetuning.
> What we use in practice:    CFG pretraining **+** CCA finetuning.  (Because we have no resources to pretrain visual AR models ourselves. The unconditional part of existing models is wasted/useless)
>
> **For 'CCA+CFG' experiments:**
> We leverage pretrained CFG model  **+**  CCA finetuning (but **Randomly dropout condition labels with 10% probability**).
>
> 'CCA only' Method does not require CFG pretraining at all.  CCA+CFG requires so, but is only related to Section 5.4 of the paper, which allows us to outperform CFG by integrating CFG into CCA.
>
> **Q3: In Sec 5.4,  certain unconditional data samples are used as positives but it is not clear how exactly that works. "**
>
> **A3:**
> We have added Pseudo code in Appendix D to facilitate understanding of our method and thank the review for raising this issue~
>
> Sec 5.4 is only related to the CCA+CFG algorithm, which requires unconditional models to be additionally trained compared with 'CCA only' method.
>
> In simple words, during training:
> 1. We sample 256 images $[x_i]$ and their labels $[c_i]$ from ImageNet.  Then we randomly shuffle $[c_i]$ to form negative labels $[c_i^{\text{neg}}]$.
> 2. We concat $x = [x_i, x_i]$ and $c=[c_i, c_i^{\text{neg}}]$. This results in a training batch $[x,c]$ of 512 data including both postive and negative data.
> 3. **If CCA+CFG training (sec 5.4).**  We randomly select 10% labels from $[c_i, c_i^{\text{neg}}]$ and replace them with unconditional [MASK] tokens.
> 4. According to Eq 12 in the paper. For each data within 512 training batch: $Loss =  - \log \sigma[ \beta \log \frac{p_\theta}{p_\phi} (x|c)]$ if $c$ is the positive label for $x$, **or if $c$ is unconditional [MASK]**.  Otherwise, we have $Loss =  - \lambda \log \sigma[ - \beta \log \frac{p_\theta}{p_\phi} (x|c)]$ if $c$ is negative label.
>
> The loss for unconditional data is the same as the loss for positive data. (Basically, you can consider it as Maximum likelihood (MLE) Loss because it always increases likelihood $p_\theta(x|c=[MASK])$. )
>
> **Q4: Fig. 2 has no verbal explanation (Uninformative)."**
>
> **A4:**
> We have added verbal explanations of Figure 2 to facilitate understanding. We thank the reviewer for the detailed suggestion.

---

> ### Author Response · Authors · 2024-11-18
> **Official Response to Reviewer 8sAn (Part 2/2)**
>
> **Q5: I wonder if proposed method can only work when the training batch size is large enough, because CCA resembles CLIP-like contrastive loss.**
>
> **A5:** **We are very certain CCA does not require a large batch size.**
>
> For CCA, we conduct all experiments using a batch size of **256** in our paper.  For reference,  CLIP uses a batch size of 32768 in their work.  LlamaGen visual AR pretraining uses batch size 768.
>
> CCA even works well with a batch size of **32**.
>
> |    | LlamaGen-L | +CCA (bz 256) | +CCA (bz 32) |
> |----|---|---|---|
> | IS  | 64.3| 288.2 | 269.9 |
> | FID | 19.07| 3.43 | 3.52 |
>
> We do appreciate the reviewer for bringing up this insightful question. We have some guesses on the distinctive feature between CCA and CLIP:
>
> **Insight:** CLIP is more like DPO. They target optimizing **relative** values across various data points. This kind of method requires we compare **many** data points within the same training batch in order to be stable. This might exactly be the reason explaining why DPO significantly fail in our experiments in Sec 5.3 (batch size 256 is too small). In contrast, CCA optimizes **absolute** likelihood of a single data point, so it is much more stable and does not rely on a very large batch size.

---

> ### Author Response · Authors · 2024-11-20
> **Further response to Reviewer 8sAn (Part 3/3)**
>
> We thank the reviewer for the prompt feedback and are glad to see our response helps.
>
> **Q6: When CCA is combined with CFG, IS becomes worse while FID becomes better. Can the authors explain this phenomenon?**
>
> **A6:** Sorry we miss this question in our initial response.
>
> Roughly, as indicated by Figure 6 in the paper:
>
> For IS:   CCA >  CFG+CCA > CFG
>
> For FID:  CFG+CCA > CFG > CCA  ( ">" means better)
>
> Frankly speaking, we are not **entirely** sure why this happens. We think/guess this is mainly related to hyperparameter choices, particularly $\lambda$ (Figure 5). We all know that there is a trade-off between IS and FID. For CCA-only experiments, we usually choose a larger $\lambda$ than that chosen by CCA+CFG experiments (Figure 5, right 1). A larger $\lambda$ can effectively increase the IS score  (Figure 5, left 2). Although for CCA+CFG models, we can also increase IS score from 260 to 300+ by increasing the guidance scale $s$, we did not do so in our paper, because it will harm FID score a little bit.
>
> Overall, all models are selected to have minimal FID. The IS is not a main focus in our hyperparameter selection. This may cause IS score to have some "unusual behavior".
>
>
> **Q7: For tasks that do not rely on class labels like T2I that rely on captions, would your method still generalize well and work similarly as the ImageNet case using random shuffling? Or would you need another strategy?**
>
> **A7:** I believe CCA should generalize/apply well on T2I models or text labels. No other strategy should be needed.
>
> First, theoretically speaking, CCA can be **directly** applied to T2I models because, similar to image labels, text labels can also be sampled and shuffled in the same way. (We designed this algorithm to be compatible with text settings right from the beginning.)
>
> If CCA is applied to T2I models, I cannot see any reason why CCA should succeed on C2I models, but fail on T2I models. I mean, they are all conditions. The sole difference is that there are finite class labels, but infinite text labels. This makes no difference to CCA because it does NOT require exhaustively traversing all possible labels.
>
> We understand it's best to give some experimental evidence on T2I models here. However, there currently lacks a publicly accessible pretrained foundational T2I visual AR model on **public datasets** like LAION-5B and we lack enough computing resources in limited rebuttal time. We thus hope the reviewer finds it acceptable that we only argue this point verbally.
>
> ***
> We'd be grateful if the reviewer could consider re-evaluating our work based on our additional input.

---

### Official Review · Reviewer_WGPM · 2024-11-04

**Soundness:** 3
**Presentation:** 2
**Contribution:** 3
**Rating:** 8
**Confidence:** 3

**Summary:**

This paper proposes CCA to replace CFG in auto-regressive models, which saves sampling costs while keeps the sampling performance on part with CFG. And this paper gives the theoretical connection

**Strengths:**

1. The theoretical derivation is clear and logically structured, allowing for a comprehensive understanding of the underlying concepts.
2. The authors have conducted a thorough series of experiments that effectively validate the theoretical claims made in the paper.

**Weaknesses:**

1. concerns mentioned in questions.
2. lack of explanation for notations when first mentioned, such as $p_{\theta}$  in line 168, $\sigma$ in line 189

**Questions:**

1. Your pipeline requires only 1% of the pre-trained epochs. However, could you provide insights into the corresponding GPU memory usage during training in comparison to CFG?
2. Regarding the proposed loss function in Equation 11, what is its convergence speed?
3. A more detailed inquiry: in line 372, the experiments do not utilize a common default value of s=7.5, for Classifier-Free Guidance (CFG). Could you elaborate on the reason behind this choice?

---

> ### Author Response · Authors · 2024-11-18
> **Official Response to Reviewer WGPM**
>
> We thank the reviewer for the valuable comments and seek to address reviewer's concerns below:
>
> **Q1: Requires only 1% epochs.... However, could you provide insights into the corresponding GPU memory usage during training in comparison to CFG?"**
>
> **A1:**
> Take LlamaGen-L model as an example. Say batch size 256. CFG training takes about **32 GB** GPU memory and CCA requires **60 GB**. This is mainly because CCA requires comparing positive and negative data, resulting in two times of batch size. **This is similar to all mainstream alignment algorithms such as DPO.**
>
> **Q2: What is its convergence speed regarding the loss function in Equation 11. (CCA loss)"**
>
> **A2:** Our initial experiments on both LlamaGen and VAR show that the method quickly converges within 1 epoch, (~5000 gradient steps). Additional training does not bring much benefit regarding FID. We rerun an experiment on LlamaGen-L and here is the result
>
> | Training Epochs| 0 epoch| 0.2 epoch|0.4 epoch|0.6 epoch|0.8 epoch| 1 epoch (Chosen) | 3 epoch|
> |-------|------|-----|-----|----------|-----|----------|-----|
> | FID            | 19.1 | 4.71| 3.96 | 3.83|  3.54         | 3.51 | 3.49|
>
>
> **Q3: In line 372, the experiments do not utilize a common default value of s=7.5 for CFG. Could you elaborate on the reason behind this choice?"**
>
> **A3:** We choose $s=3$ for comparison because this is **exactly the default/recommended value** of the LlamaGen  model in their official code base. [1]  For VAR the officially recommended value is 4.0 [2].  $s=7.5$ might be more suitable for diffusion models. but **for VAR/LlamaGen $s=7.5$ would basically not increase image quality and will hurt image diversity.**
>
> Just to be sure we compare with s=7.5 VAR samples and s=3 samples in the link below (14*9 pairs wo/ cherry picking) :
> https://anonymous.4open.science/r/ICLR2025rebuttal-9DD3
>
> [1] https://github.com/FoundationVision/LlamaGen/blob/ce98ec41803a74a90ce68c40ababa9eaeffeb4ec/autoregressive/sample/sample_c2i.py#L117  section 2. Line 7. (Please note that their s=4 is exactly our s=3, because our paper uses a slightly different definition for s compared with LlamaGen .)
>
> [2] https://github.com/FoundationVision/VAR/blob/main/demo_sample.ipynb
>
>
> **Q4: Lack of explanation for notations when first mentioned, such as $p_\theta$ in line 168, $\sigma$ in line 189.**
>
> **A4:** We thank the reviewer for the detailed suggestion. All mathematical notations should be described by text when first referenced. We have revisited the paper to ensure this does not happen again.
>
> $\sigma(\cdot)$ is the standard logistic (sigmoid) function: $\sigma(w) := 1/ (1+e^{-w})$.
>
>
> | Distributions |$p(x)$ | $p(x\mid c)$ |  $p^{\text{sample}}(x\mid c) \propto p(x\mid c) \left[\frac{p(x\mid c)}{p(x)}\right]^s$ |
> |--|--|--|--|
> | Explanation          | Data Unconditional Dist.      | Data Conditional Dist.      | Our target sampling Dist.    |
> | Models               | $p_\phi(x)$                                | $p_\phi(x\mid c)$                              | $p_\theta^{\text{sample}}(x\mid c)$  |

---

> > ### Comment · Reviewer_WGPM · 2024-11-23
> > **Thanks for responses**
> >
> > Thanks for your response. I have no other concerns

---

> > > ### Author Response · Authors · 2024-11-23
> > >
> > > Thank you for the feedback. We are glad that our responses help.

---

### Official Review · Reviewer_bJkX · 2024-11-04

**Soundness:** 3
**Presentation:** 3
**Contribution:** 3
**Rating:** 6
**Confidence:** 3

**Summary:**

To bridge the gap between autoregressive language and visual generation, this work targets at removing the need of CFG in AR image generation, therefore unifying the sampling scheme and reducing the sampling cost. The authors choose to fine-tuning the pretrained AR models with the contrastive alignment loss. The intuition behind the contrastive alignment is to maximize the likelihood of positive conditions and minimize the likelihood of the negative conditions. More discussion with classifier guidance and classifier-free guidance is also illustrated, providing more insights for difference sampling schemes. Experiments are conducted on autoregressive visual generation methods like LlamaGen and VAR.

**Strengths:**

* The problem of sampling schemes for autoregressive models is fundamental and should be explored.

* Detailed illustrations are provided. For example, the authors review preliminaries, give formal derivation, and also the practical implementation.

* It is interesting to see that CCA provides extra gain based on CFG.

**Weaknesses:**

* For the motivation, one confusion to me is why removing the CFG is necessary for unifying visual and language generation. I mean, there may no influence for sampling them with slightly different schemes? I recognize that introducing CCA can reduce the half cost. But it is achieved at the cost of sampling quality as demonstrated in Table 2.

* To my understanding, the model should be fine-tuned for each lambda and beta individually. Is that correct? If so, this will be a big limitation since other sampling schemes can adjust trade-offs during inference, which is more flexible.

**Questions:**

Could the authors clarify the necessity of removing CFG, except sampling cost?

Should the model be fine-tuned individually for each lambda and beta?

---

> ### Author Response · Authors · 2024-11-18
> **Official Response to Reviewer bJkX (Part 1/2)**
>
> We thank the reviewer for the valuable comments and seek to address reviewer's concerns below:
>
> **Q1: Motivation: Could the authors clarify the necessity of removing CFG, except sampling cost?? Having different sampling scheme for vision and language seems acceptable (good enough).**
>
> **A1:** Sampling cost is actually only a small part of our initial motivation. We are strongly motivated mainly because:
>
> 1. **Unified models SHOULD have unified algorithms.** If we've already got an AR model that can both output text and images (like Emu3). Let's say we are happy with image sampling with CFG while text sampling without CFG. Instead, we improve text quality by applying alignment algorithms (preference training).  Now, what if we suddenly want to include speech/audio data into our unified AR model? After we tokenize audio data, the dilemma we face is whether we should choose CFG or alignment algorithms to improve its sampling quality? What about 3D data and graph structure data?
>
> The point I'm trying to make above is that for **unified** AR models, **we really shouldn't give a special trick to a certain group of tokens just because they correspond to some special content** (visual rather than text). CFG is acceptable. However, it is simply not good enough because it makes our models seemingly less "unified". What we want is let tokenizers take care of various-modal data, and let AR models only worry about modeling these tokens. Modular everything, no inductive bias. Looking at the bigger picture, **Consistency is what this paper cares about.**.
>
> 2.  **Training Inconvenience.** CFG is all fine when training pure text2image AR models. However, when training multi-modal AR models, it becomes a bit tricky: Our data could be [text1] [image] [text2] and we want to jointly optimize prediction loss for both [text1-2] and [image] with a singular loss. However, CFG requires randomly masking image conditions [text1] during training. When we do so, our data becomes [MASK] [image] [text2]. This way we can only have a meaningful loss for [image]. The prediction loss for [MASK] and [text2] is now meaningless. This is kind of annoying.
>
> 3. **Alignment Inconvenience.** Language models require alignment procedure anyway. However, if our AR models use CFG. When later performing DPO, how do we calculate the model likelihood for image tokens? Do we also need to consider CFG by integrating unconditional likelihood for the reference model?
>
> 4. **Theoretical meaning**. It makes no sense that alignment methods work well with language data but not with visual data.  In our paper sec. 4,  we show it targets the same thing as CFG. The work should be meaningful even if it does not eventually replace CFG.
>
> **Q2: CCA can reduce the half cost. But it is achieved at the cost of sampling quality (Table 2).**
>
> **A2:** We agree with the reviewer that the performance of 'CCA only' somewhat lags behind CFG method currently. However, CCA is still quite an explorative work in the current stage, we believe **its main contribution is pointing a promising possible direction for future visual alignment works** (the very first work to our knowledge). After all, we successfully **lower the FID from 9.38 to 2.69 with one simple loss function** for a 3B model. This is 93% closer to the CFG performance. We are actively working on closing the final 7% gap.  We would be grateful if the reviewer finds this performance acceptable and indicative.
>
> Additionally, "CFG+CCA" method clearly outperforms CFG in all dimensions in our experiments (Figure 6), though somewhat marginally.

---

> > ### Comment · Reviewer_bJkX · 2024-11-26
> > **Thanks for the response**
> >
> > I have thoroughly reviewed the authors' response and appreciate the comprehensive explanation provided. I concur that the submission meets the standard for acceptance at ICLR. Considering the limitations, I prefer to retain my score of 6.

---

> > > ### Author Response · Authors · 2024-11-26
> > >
> > > Thank you for the feedback and valuable suggestions. We will continuously improve our work.

---

> ### Author Response · Authors · 2024-11-18
> **Official Response to Reviewer bJkX (Part 2/2)**
>
> **Q3: The model should be fine-tuned for each lambda and beta individually? This will be a big limitation since other sampling schemes can adjust trade-offs during inference, which is more flexible.**
>
> **A3:** Yes, in our original paper, the model is finetuned for each hyperparameter. This is indeed a limitation for CCA. Nonetheless, **we think this is not a *very* BIG concern due to the following reasons**:
>
> 1. We find $\beta$ and $\lambda$ affect CCA performance similarly (Appendix C in the paper). **It's often not necessary to tune them both if not for academic research**. We recommend fixing $\beta$ and adjusting $\lambda$ only.
> 2. We can make $\lambda$ as input of our model, and randomly sample $\lambda \in [e^0, e^{8}\approx3000]$ during training. This way we can train models with multiple $\lambda$ simultaneously and only test out optimal hyperparameters during inference, **directly solving the above problem.** A similar method has been used by [1].
>
> New experimental result: Flexible $\lambda$ as input (only 1 model, 3 epochs training):
> | Inference-time $\lambda$  |   Base Model    | 10     | 100   | 300 (Chosen) | 1000  | 3000  | 10000 |
> |------------|-------------|--------|-------|--------------|-------|-------|-------|
> | FID        | 19.07       | 7.23   | 4.18  | **3.59**         | 3.73  | 4.12  | 5.33  |
> | IS         | 64.3        | 153.1  | 218.8 | **256.2**        | 277.5 | 307.7 | 341.2 |
>
> (Please note that the above results are very initial and without any tuning/ablation)
>
> 3. Although for academic research we want this flexibility. **During real-world deployment flexibility is often not necessary**, most of the time we want the best model performance for users instead of letting users find out which hyperparameter is the best. LLM alignment similarly requires tuning alignment parameters like $\beta$ and this is often not a concern.
> 4. **CCA is extremely efficient.** Even if we tested 10 points to get a final satisfying model. That is **only 10% of computation/time compared with pretraining.**
>
> [1] On Distillation of Guided Diffusion Models. Chenlin Meng, Robin Rombach, Ruiqi Gao, Diederik Kingma, Stefano Ermon, Jonathan Ho, Tim Salimans. CVPR 2023.

---

> > ### Author Response · Authors · 2024-11-25
> > **Looking forward to your feedback**
> >
> > Dear reviewer,
> >
> > Thank you for your valuable comments regarding our submission.
> >
> > We have posted our responses and hope to address your concerns about the motivation for removing CFG in AR models besides computational cost. We also explain why CCA can also be flexible like CFG by allowing diversity-fidelity trade-offs at inference time.
> >
> > Could you please take a look at our responses and possibly re-evaluate our work based on the additional input? Thank you!
> >
> > Best regards,
> >
> > The Authors of the paper: Toward Guidance-Free AR Visual Generation via Condition Contrastive Alignment

---

### Author Response · Authors · 2024-11-18
**Review and Rebuttal Summary**

We would like to thank all the reviewers for their time and valuable comments.

We are happy to see all six reviewers recognize that our method is theoretically grounded and has good contributions. Reviewer Zf6F, 8sAn, CPMH and qqAa think our paper proposes an important and fundamental problem (removing CFG from AR models).  Reviewer qqAa, 8sAn, CPMH, and Zf6F find the performance gain is significant, and experimental results are promising. Reviewer Zf6F and CPMH think the proposed method is simple and elegant.

Main concerns about our work are 1. CCA somewhat lacks flexibility compared with CFG because it requires training multiple models in order to find optimal hyperparameters. 2. CCA currently still lags a little behind CFG regarding performance. 3. Some implementation details are unclear, especially regarding the CCA+CFG algorithm proposed in the experimental section.

We summarize the main actions taken during the rebuttal:
1. We conduct new experiments to show that CCA can gain inference-time flexibility just like CFG, by conditioning the model on some training parameters and randomly sampling these parameters during training. (Appendix E)
2. Experiments to study the convergence speed and stability of CCA.  (Appendix C)
3. Experiments to compare batch size and computational cost/ GPU memory requirement. (Appendix C)
4. Add pseudo code and verbal explanations in the paper to better reveal experimental details. (Appendix D)
5. Clarify the motivation of our paper and several confusions or misunderstandings regarding our paper.

We look forward to further discussions with the reviewers!

---

### Meta-Review · Area_Chair_dKGZ · 2024-12-20

**Metareview:**

CFG improves visual generative models but creates inconsistencies in autoregressive (AR) multi-modal generation. To address this, this paper proposes Condition Contrastive Alignment (CCA), which directly fine-tunes pre-trained models to achieve guidance-free AR visual generation. CCA significantly enhances model performance with minimal fine-tuning, reducing sampling costs by half while maintaining a balance between sample diversity and fidelity.

All reviewers unanimously agree to accept it.

Though limitations such as models need to be fine-tuned with each parameter exist, this paper could still be widely interesting to the whole visual generation community. Given the current promising results presented in this paper, I recommend to accept it.

**Additional Comments On Reviewer Discussion:**

Initial concerns include: the model should be fine-tuned for each lambda and beta individually; not sure about the effectiveness with small batch size; some clarity issues; missing implementation details. After rebuttal, authors fixed most of them while also admit the existing limitations.

Some open questions include whether this approach can improve MaskGIT style models.

---

### Decision · Program_Chairs · 2025-01-22

Accept (Oral)